# The AM-4 Family of Layered Titanosilicates: Single-Crystal-to-Single-Crystal Transformation, Synthesis and Ionic Conductivity

**DOI:** 10.3390/ma17010111

**Published:** 2023-12-25

**Authors:** Galina O. Kalashnikova, Sergey V. Krivovichev, Victor N. Yakovenchuk, Ekaterina A. Selivanova, Margarita S. Avdontceva, Gregory Yu. Ivanyuk, Yakov A. Pakhomovsky, Darya V. Gryaznova, Natalya A. Kabanova, Yelizaveta A. Morkhova, Olga Yu. Sinel’shchikova, Vladimir N. Bocharov, Anatoly I. Nikolaev, Olga F. Goychuk, Sergei N. Volkov, Taras L. Panikorovskii

**Affiliations:** 1Laboratory for Synthesis and Research of the Properties of Mineral-Like Functional Materials, Nanomaterial Research Center of the Kola Science Centre, Russian Academy of Sciences, Fersmana str. 14, 184209 Apatity, Russia; g.kalashnikova@ksc.ru (G.O.K.); s.krivovichev@ksc.ru (S.V.K.); v.yakovenchuk@ksc.ru (V.N.Y.); selivanova@geoksc.apatity.ru (E.A.S.); g.ivanyuk@ksc.ru (G.Y.I.); pakhom@geoksc.apatity.ru (Y.A.P.); d.gryaznova@ksc.ru (D.V.G.); a.nikolaev@ksc.ru (A.I.N.); o.goychuk@ksc.ru (O.F.G.); 2Department of Crystallography, Institute of Earth Sciences, St. Petersburg State University, 7–9 University Emb., 199034 St. Petersburg, Russia; m.avdontceva@spbu.ru; 3Geological Institute, Kola Science Center of Russian Academy of Sciences, 14 Fersman Street, 184200 Apatity, Russia; 4Laboratory of Nature-Inspired Technologies and Environmental Safety of the Arctic, Nanomaterial Research Center of the Kola Science Centre, Russian Academy of Sciences, Fersmana str. 14, 184209 Apatity, Russia; n.kabanova@ksc.ru; 5Samara Center for Theoretical Materials Science, Samara State Technical University, Molodogvardeyskaya Str. 244, 443100 Samara, Russia; 6Institute of Experimental Medicine and Biotechnology, Samara State Medical University, Chapayevskaya Srt. 89, 443099 Samara, Russia; eliztimofeeva@mail.ru; 7Laboratory of Physicochemical Design and Synthesis of Functional Materials, Institute of Silicate Chemistry of Russian Academy of Sciences, 2 Adm. Makarova, 199034 St. Petersburg, Russia; sinelshikova@mail.ru; 8Geo Environmental Centre “Geomodel”, St. Petersburg State University, Universitetskaya nab., 7/9, 199034 St. Petersburg, Russia; bocharov@molsp.phys.spbu.ru; 9Tananaev Institute of Chemistry of the Kola Science Centre, Russian Academy of Sciences, Academic Town, 26a, 184209 Apatity, Russia; 10Laboratory of Arctic Mineralogy and Material Sciences, Nanomaterial Research Center of the Kola Science Centre, Russian Academy of Sciences, Fersmana str. 14, 184209 Apatity, Russia; s.volkov@ksc.ru

**Keywords:** mineral-mimetic material, SCSC, AM-4, lintisite, kukisvumite, L3, K3, transformation, microporous, arctic minerals, ionic conductivity, ion-migration modeling, impedance spectroscopy

## Abstract

Flexible crystal structures, which exhibit single-crystal-to-single-crystal (SCSC) transformations, are attracting attention in many applied aspects: magnetic switches, catalysis, ferroelectrics and sorption. Acid treatment for titanosilicate material AM-4 and natural compounds with the same structures led to SCSC transformation by loss Na^+^, Li^+^ and Zn^2+^ cations with large structural changes (20% of the unit-cell volume). The conservation of crystallinity through complex transformation is possible due to the formation of a strong hydrogen bonding system. The mechanism of transformation has been characterized using single-crystal X-ray diffraction analysis, powder diffraction, Rietvield refinement, Raman spectroscopy and electron microscopy. The low migration energy of cations in the considered materials is confirmed using bond-valence and density functional theory calculations, and the ion conductivity of the AM-4 family’s materials has been experimentally verified.

## 1. Introduction

The detailed study of the crystal structures and physical and chemical properties of mineral-mimetic materials with functional properties is of interest from the viewpoint of material sciences, in order to improve the methods of obtaining synthetic mineral analogues and to determine the possibility of their use in industry and agriculture [1,2]. The level of human environmental impact is extremely high and the problem of treating industrial waste containing radionuclides and heavy nonferrous metals is as acute as ever. Ion exchangers are of special interest, especially meso- and microporous zircono-, niobo- and titanosilicates, which in many respects are similar to zeolites, but are more stable in aggressive media [3]. Such materials are already widely used in industry for applications including extracting radionuclides from liquid radioactive waste (LRW) [4,5,6,7,8], gas separation [9,10,11], water treatment [12], as catalysts [13,14,15], nanofiber filters, mineral−organic nanocomposites for drug delivery, bio-sensors, optical filters, materials for electronics, mechanics and other purposes [16,17].

The significant diversity of zircono-, niobo- and titanosilicates in nature is typical for alkaline complexes (Khibiny, Lovozero, Kovdor and Murun in Russia, Saint-Hilaire in Canada, Ilimaussak in Greenland, Aris in Namibia, etc.). The massifs of phoidolites and nepheline syenites in Khibiny and Lovozero are the world’s largest, and are the main “suppliers” for the new minerals that serve as prototypes for creating new materials with functional properties [18,19,20,21,22]. The most representative mineral-mimetic materials are the following: Engelhard Titanium Silicate-4 (ETS-4) molecular sieve (analogue of zorite [23]) [24], the base for antimicrobial drugs [25]; Aveiro-Manchester-4 titanosilicate (Na-analogue of kukisvumite [26]) (AM-4) [27]; selective sorbent Cs from LRW analog of sitinakite [28] IONSIV IE-911 [29,30] and analogue of ivanyukite-Na [6,31]—synthetic ivanyukite (SIV) [32,33], one of the most effective sorbents of Cs, Sr, Co, Tl, Pb, Ag and REE from aqueous solutions [34,35,36,37].

Natural materials related to the AM-4 family belong to the five mineral species listed in Table 1. Minerals of the lintisite-kukisvumite group crystallize at the late-stage of hydrothermal activity as a result of alteration of earlier titanosilicates: lorenzenite, lamprophyllite, lomonosovite and murmanite [38]. All minerals of the lintisite-kukisvumite group, excluding punkaruaivite, contain a significant amount of Na. The Na-containing mineral-mimetic materials are of interest due to their potential Na conductivity. It is noteworthy that many of the existing promising fast Na^+^-ion conductors have mineral prototypes with the crystal structures of perovskite [39], alluaudite [40] and brucite [41]. One of the most effective conductors is NASICON [42], which is related to the zirconosilicates and potential mineral species that might be found in alkaline complexes in associations similar to those of the lintisite-kukisvumite-group minerals.

AM-4 demonstrates excellent sorption properties with respect to Pb from industrial solutions, but the molecular mechanism of this process remains unexplored [47]. It has also been observed that, in acidic environments, AM-4 loses extra-framework Na cations and H_2_O molecules, with the acid-activated form being more efficient with respect to the adsorption of N_2_ and CO_2_ [48]. The acidic form of AM-4 can be used as a renewable sorbent for the Ag and I uptakes, with at least five cycles of transformation without loss of sorption properties [49]. Acid modification of AM-4 with HNO_3_ is used for the reaction of catalysis cyclocondansation between 1,2-phenylenediamine and acetone [13] and transesterification of glycerol with dimethylcarbonate [50]. The XRD powder pattern of the acid-modified form significantly differs from the initial AM-4 [13]; the crystal structure of this phase is unknown. Lintisite and kukisvumite, like their synthetic Na analog AM-4, have weak exchange properties for Cs, Rb and K [51]. It was also found that, in acidic environments, lintisite and kukisvumite quickly lose not only extra-framework Na cations and water molecules, but also “cross-linking” Zn or Li cations turning into the materials L3 and K3. The L3 and K3 designations denote compounds obtained via the treatment of kukisvumite and lintisite during 3 h with 0.5 M hydrochloric acid [51]. The L3 material has the same powder pattern as the H-exchanged form of AM-4, whereas the powder diffraction pattern of K3 slightly differs from that of L3 [13,49,51].

Herein, we report the mechanism of a single-crystal-to-single-crystal transformation from natural crystals of kukisvumite and lintisite into crystals of the K3 and L3 forms, respectively, with the details of their crystal structures and chemical composition. The topology and the cation-migration paths’ compounds in the AM-4 family compounds were analyzed theoretically using a geometrical–topological (GT) approach [52]. The data on the conductive properties were also obtained using bond valence site energy (BVSE), kinetic Monte Carlo (KMC) and density functional theory (DFT) calculations, compared with the electrochemical data obtained using the impedance spectroscopy.

## 2. Materials and Methods

### 2.1. Mineral Samples

The samples of kukisvumite (white needle-like crystals or radial and sheaf-like aggregates) studied in this work (Figure 1a) were taken from lamprophyllite-aegyrine-microcline pegmatite in the urtites of Kukisvumchorr Mt. (Khibiny massif). The rims of the kukisvumite crystals are in some places significantly enriched in manganese up to the transition into manganokukisvumite (Figure 1d). In the marginal part of the pegmatite, small needle-like pseudomorphs of kukisvumite form after plates of lamprophyllite, in the intermediate part the mineral forms intergrowths with aegirine, whereas in the central part kukisvumite forms monomineralic nests up to 8 cm in diameter.

The lintisite samples used in this study were taken from ussingite-aegirine-microcline pegmatite, occurring in the contact zone of foyaites and poikilite sodalite-nepheline syenites of Alluaiv Mt. (Lovozero massif), where it forms partial pseudomorphoses upon lorenzenite, separate colorless needle-like crystals up to 1.5 cm in length (Figure 1c) and their parallel sheaf-like aggregates up to 4 mm in cross section in the voids of pegmatite polymineralic zone.

Punkaruaivite (Figure 1b) is a Na-free analogue of lintisite discovered in the natrolite-microcline pegmatite in the foyaites of the Eveslogchorr Mt. (Khibiny massif) and in the ussingite-egirine-microcline pegmatite in the rocks of the stratified complex of the Maly Punkaruaiv Mt. (Lovozero massif).

Eliseevite (Figure 1e) is a cation-deficient analogue of lintisite discovered in the ussingite-aegirine-microcline pegmatite Maly Punkaruaiv Mt. and the Microcline-Sodalite-Ussingit pegmatite Alluaiv Mt. in rocks of the Lovozero stratified complex.

### 2.2. Synthesis

Reagents used were of reagent or analytical-grade quality, obtained from commercial suppliers and used without further purification (Merck, Aldrich and Neva reactive). The titanium ammonium sulfate (NH_4_)_2_TiO(SO_4_)_2_∙H_2_O (STA) is the product of titanite ore treatment (mining JSC «Apatit», PhosAgro company (Moscow, Russia) and Kola Science Centre, Russia).

The hydrothermal synthesis of AM-4 based on the STA salt was performed as reported previously [27,49]. The mixture of 17.6 g H_2_O, 5.024 g Na_2_SiO_3_∙5H_2_O and 2.89 g STA or 5.95 g TiCl_3_ was used for the preparation of the Li modification of AM-4 (Li-AM-4). For this experiment, NaOH was replaced by LiOH (1.6 g). The process of the synthesis was carried out in the stainless autoclaves with PTFE liners for hydrothermal synthesis (TOPH, Seoul, Republic of Korea); the volume of each autoclave was 25 mL. The synthesis proceeded with stepwise temperature gradients (100 °C, 150 °C and 230 °C) according to the original method without forced stirring [53].

### 2.3. Acidic Treatment

Several crystals of natural kukisvumite and lintisite have been kept in 0.5 M HCl for 3 h with periodic shaking of the vessels. After the removal of acid solution, the crystals were washed with distilled water and dried in air. The crystals did not change their shape or color (Figure 2), which indicated that transformation is of the single-crystal-to-single-crystal (SCSC) character, confirmed by single-crystal X-ray diffraction analysis. The samples of kukisvumite and lintisite subjected to 3 h treatment in 0.5 M HCl are denoted as K3 and L3, respectively.

### 2.4. Chemical Analysis

The morphological characterization of the synthetic powders of titanosilicates (Figure 1f) was carried out using a scanning electron microscope LEO-1450 (Carl Zeiss Microscopy, Oberkochen, Germany). The chemical composition of the synthetic products was studied with an Oxford Instruments Ultim Max 100 analyzer at 20 kV, 500–1000 pA, 1–3 µm beam diameter (Geological Institute of FRC KSC RAS).

The chemical composition of mineral samples has been studied by wavelength-dispersive spectrometry using a Cameca MS-46 electron microprobe (Geological Institute, Kola Science Centre of the Russian Academy of Sciences, Apatity) operating at 20 kV, 20–30 nA and 5 μm beam diameter.

### 2.5. Powder and Single-Crystal X-ray Diffraction

Theoretical powder diffraction pattern for the acidic form of AM-4 (L3) was calculated on the basis of crystal structure data using the VESTA 3 program [54] for (λ_1_ = 1.54059 and λ_2_ = 1.54432 Å) using RIETAN-FP algorithms [55], based on the atomic coordinates and unit-cell parameters.

Powder X-ray diffraction data (PXRD) collected by means of a Rigaku «MiniFlex II» diffractometer in the 2θ range of 5–60° (Cu*K*α; 1.5418 Å) with scanning steps of 0.02° in 2θ. The normal-focus Cu X-ray tube was operated at 30 kV and 15 mA. The unit-cell parameters were calculated using a unit-cell program [56].

Single-crystal X-ray diffraction study of kukisvumite, lintisite, K3 and L3 was performed at the Centre of the Collective Use of Equipment, Kola Science Centre using Rigaku XtaLAB Synergy-S diffractometer equipped with a hybrid photon counting detector HyPix-6000HE with monochromatic Mo*K*α radiation (λ = 0.71069 Å) at room temperature. More than half of the diffraction sphere was collected with scanning step 1° and exposure time 10–100 s. The unit-cell dimensions were refined using least-squares techniques. The data were integrated and corrected by means of the CrysAlisPro program package (v41.104a), which was also used to apply empirical absorption correction using spherical harmonics, implemented in the SCALE3 ABSPACK scaling algorithm [57]. The structure was solved using SHELXT and refined via SHELXL software package (v2014/6) [58]. The positions of the hydrogen atoms in the crystal structure of lintisite were located using difference Fourier maps. The crystal structures of kukisvumite and lintisite are in a good agreement with the previously published data [59,60].

Reconstruction of the sections of the reciprocal diffraction space (Figure 3) demonstrates that crystals of kukisvumite subjected to acid treatment lost their crystallinity very significantly. Whereas the (*h*0*l*) section (Figure 3a,c) has a reasonably good quality of diffraction spots in both phases, reflections in the (0*kl*) section (Figure 3b,d) are smeared and possess many diffuse features in K3 that indicate that the phase transformation involved structural reconstructions within the *bc* plane. Due to the low quality of crystals, the structural refinement was far from perfect and was allowed to decipher basic structural features only; thus, the obtained data can be considered as a general structural model and agrees well with the powder XRD data.

The SCXRD data are deposited in CCDC under entries No. 2266154–2266157. Crystal data, data collection information and refinement details are given in Table 2. Atom coordinates and isotropic parameters of atomic displacements are given in Appendix A, interatomic distances in Appendix A and the anisotropic parameters of atomic displacements are given in Appendix A.

### 2.6. Rietveld Refinement

For the Rietveld refinement, PXRD patterns were collected on Rigaku SmartLab SE (Rigaku Corporation, Tokyo, Japan) 3 kW sealed X-ray tube, D/teX Ultra 250 silicon strip detector, vertical type θ-θ geometry, HyPix-400 (2D HPAD) detector. PXRD data were collected at room temperature in the 2θ range between 3° and 120° with a step interval of 0.02°. Rietveld refinement was performed on the powder diffraction patterns. The structure models of Ti(Si_2_O_5_(OH))(OH) were used as starting models, obtained from SCXRD (without H atoms), in the refinement utilizing RietveldToTensor software (v1.1) [1].

### 2.7. Raman Spectroscopy

The Raman spectra (RS) of kukisvumite and K3 (after 40 min in 0.1 M solution of HNO_3_ acid) collected from uncoated individual grains were recorded with a Horiba Jobin-Yvon LabRAM HR800 spectrometer equipped with an Olympus BX-41 microscope in backscattering geometry (Saint-Petersburg State University). Raman spectra were excited by a solid-state laser (532 nm) with actual power of 2 mW under the 50× objective (NA 0.75). The spectra were obtained in the range of 70–1300 cm^–1^ at the resolution of 2 cm^−1^ at room temperature. To improve the signal-to-noise ratio, the number of acquisitions was set to 15. The spectra were processed using the algorithms implemented in Labspec v3.15 and OriginPro v8.1 software packages.

### 2.8. Theoretical Studies of Topology and Conductivity Properties

A large number of ion conductors, metal–organic frameworks, zeolites and intermetallides have been studied using GT analysis [61,62,63,64]. The most common calculation procedures are automatic and implemented in ToposPro [65] (http://topospro.com (accessed on 30 June 2023)) or service TopCryst (http://topcryst.com (accessed on 30 June 2023)). We used the tiling method for the investigation of ion migration in the lintisite-group minerals [66,67].

The BVSE method was used for the calculation of the migration energies (*E_m_*) of all cations in the minerals, in order to determine the type of the working ion. This approach is implemented in the GUI-version of the softBV program [68,69]. The main idea is to calculate the deviation of the bond valence sum (BVS) from the ion charge, which should not exceed 15% for the migration of the working ion.

The ionic mobility of cations and protons was calculated using KMC simulations of supercells, with cell volumes of more than 5000 Å^3^ for (1–10) million KMC steps at the temperatures 300 K, 400 K, 500 K, 600 K, 700 K and 800 K. The KMC algorithm, which is also implemented in softBV (v2.0) [68,69], is based on the approximate site and migration energies derived from the BVSE analysis.

The migration energy barriers *E_m_* for AM-4:Li modification were also calculated using the DFT-NEB method [70,71], based on the generalized gradient approximation (GGA) with the Perdew–Burke–Ernzerhof (PBE) functional [72] and projector-augmented-wave (PAW) pseudopotentials [73], as implemented in the Vienna ab initio simulation package, VASP (v6.4.2) [74]. The cutoff energy was fixed at 600 eV in all calculations. For the geometry optimization, the convergence thresholds for the total energy and force components were chosen as 10^−6^ eV and 10^−5^ eV/Å, respectively. The shape and volume of the supercell were kept fixed at the optimized geometry. ToposPro software (v5.5.2.1) was used for the visualization of the Na^+^- and Li^+^-ion migration pathways. All NEB calculations were performed in the unit cell taken from the ICSD unit for AM-4 with the code 84261 (*A*2/*a*, *a* = 5.2012, *b* = 8.573, *c* = 29.3 Å, *α* = *β* = 90°, *ɣ* = 89.26°). This structure was used to construct a cell with the *P*1 symmetry of the composition Na_6_Si_8_Ti_4_O_27_. After cell optimization, part of the Na positions was replaced by Li, considering the fact that Na → Li was replaced during the AM-4 ion exchange in a LiCl solution. The two model cells have been constructed and optimized: Li_2_Na_4_Ti_4_Si_8_O_27_ (AM-4-I) and Li_4_Na_2_Ti_4_Si_8_O_27_ (AM-4-II). The NEB calculations for the Li^+^-ion diffusion (with fixed Na atoms) and Na^+^-ion diffusion (with fixed Li atoms) were carried separately for each model cell.

### 2.9. Electrochemical Measurements

The electrical conductivity of the lintisite can be carried out by three ions: Na^+^, Li^+^ and H^+^. The AM-4 sample containing Li^+^ cations (synthetic analog of lintisite) was prepared previously to analyze ion-conductivity properties of this compound and its Li modification. The 5 g sample of the AM-4 titanosilicate was treated with the 0.04 M LiCl solution with constant stirring (400 rpm), separated from the LiCl solution by vacuum filtration and dried at 75 °C. The AM-4:Li samples were investigated by the impedance spectroscopy methodology [75].

The hodographs of impedance (Figure 4) were prepared in the frequency range from 1 Hz to 1 MHz using impedance meter Z2000 in order to determine sample conductivity at alternating current in each measurement point. The frequency of 1 kHz at which measurements corresponded to the total resistance of the sample, without contribution of electrode processes was determined based on the graph. Additionally, the resistance of the samples was determined with a short-term (up to 5 s) application of direct current, which allows evaluation of the possible contribution of electrode processes using the difference between the two measurements.

Sample powders were first pressed into tablets with parameters *D* = 1 cm and *h* = 1.5 mm, at a press pressure of 4 tons. The electrodes (graphite and silver-containing conductive paste) were applied to the end sides of the pressed samples.

Electrical conductivity was measured for each individual sample as described below:The AM-4 sample (0.25 g) with graphite electrodes. For the impedance measuring two-contact cell, the sample was heated to 250 °C in a tubular furnace, kept at this temperature for 30 min, and resistance was recorded in the cooling mode. Then, the sample was kept at 500 °C for 30 min, after which its conductivity was recorded in the cooling mode.Two parallel AM-4 samples with silver-containing conductive paste. The samples were calcined at 550 °C for 2 h. The impedance measurements were carried out in the heating mode with thermostating at each temperature for 10–15 min. AM-4 during the measurement was heated to 723 °C. At this temperature, a sharp change in conductivity was observed; this is associated with the softening of the sample and the transition to a glassy state. To compare the conductivity of the samples, after the transition to the glassy state and this sample, a tablet was made with the following calcination mode: 550 °C for 2 h and 720 °C for 30 min, as well as additional burning of the paste at 550 °C for 1 h.The AM-4:Li sample with silver-containing conductive paste. Preparation and measurement of the electrical conductivity of the sample is similar to the examples from the second point.

## 3. Results

### 3.1. The Hydrothermal Synthesis of AM-4:Li Modification

The results of hydrothermal synthesis include standard methodologies for pure AM-4 (PXRD pattern shown in Section 3.3). However, the AM-4 material with Li synthesis contains impurities composed of other titanosilicates. It is noteworthy that the complete replacement of the LiOH reagent with LiCl leads to the formation of amorphous or undiagnosed products, while maintaining other conditions of the synthesis. There were only two points at which we identified AM-4 phase with the small sitinakite impurity. The impurity of zeolite with faujasite structure was sometimes observed. In both cases, the X-ray diffraction phase analysis of the resulting powder confirmed the presence of two phases: lintisite and unreacted Li-metasilicate (Figure 5a) (red curve) or impurity of sitinakite (Figure 5b). The estimated content of impurities does not exceed 3%. Because of the impossibility of separating the AM-4:Li phase from the accompanying sitinakite, conductivity experiments were carried out exclusively on the basis of the exchange form of AM-4 with LiCl solution.

### 3.2. Composition

The protonated form of rosette-like aggregates AM-4 (L3) demonstrates the absence of visible changes in the morphology, compared with initial AM-4 (Figure 6).

Natural lintisite contains 0.95 apfu of Li^+^ and 2.50 apfu of Na^+^, with a total charge of interlayer cations of 3.45. The H-exchanged L3 form of lintisite contains residual 0.04 apfu Na^+^, with a total interlayer charge of 0.04 (not counting H^+^ ions). Initial kukisvumite contains 0.06 apfu of Mn^2+^, 0.41 apfu of Zn^2+^ and 2.95 apfu of Na^+^, with a total charge of 3.39. The H-exchanged K3 form loses all Zn and Mn and contains residual 0.01 apfu of K^+^ and 0.01 apfu of Na^+^. The significant deficiency of the positive total charge is due to the substitution Na^+^ + O^2−^ → □ + OH^−^ and is consistent with our previous data [31]. Table 3 provides details of the analytical results for natural and acid-treated forms of lintisite and kukisvumite.

### 3.3. Acidic Treatment–Powder Diffraction

The evolution of PXRD pattern after the acid treatment leads to significant changes as demonstrated in Figure 7. The powder diffraction pattern of the initial AM-4 is in good agreement with the theoretical data and with the PDF Card No. 01-070-6975 and the same is also true for L3. The most dramatic changes are in the low-angle (5–25°) region, where the most intensive (200) reflection, at 5.98° 2θ (14.76 Å), is shifted to (100) 7.54° 2θ (11.71 Å). This shift corresponds to the contraction of the interlayer space between titanosilicate blocks in the structure. The L3 powder diffraction pattern contains an additional (110) reflection at 12.76° 2θ (6.93 Å). In the middle-angle region (25–45°) for L3, two strong reflections (12-1) 27.34° 2θ (3.25 Å) and (30-2) 38.52° 2θ (2.33 Å) appear; the intensities for the middle-range reflections (210, 011 and 21-1) are significantly higher than those for AM-4.

### 3.4. Recycling Experiments

In order to determine if the compound can be reconverted to its original structure after acid treatment, sorption properties of L3 for Na^+^ and Li^+^ cations were studied on the basis of the model solutions of 0.001 M NaOH and LiOH at 35 °C temperature and periodic shaking for 7 h. The ratio of solution and crystals was V:m = 5:0.001 (mL:g). Finally, the solid phases L3:Na and L3:Li were filtrated, washed with distilled water (25 mL) and dried in the air. The cation-exchange capacity of the L3 to Na^+^ and Li^+^ cations were 12.45 and 2.65 (mg/g), respectively.

After acid treatment, the powder X-ray diffraction pattern changed significantly. The (200) reflection at 6.02° 2θ (14.68 Å) in AM-4 after treatment shifted to (100) 7.40° 2θ (11.93 Å) in L3. Figure 8 demonstrates weak differences between L3 with L3-Li and L3-Na PXRD patterns. Subsequent treatments with LiOH and NaOH did not lead to changes in interlayer space between Ti(Si_2_O_5_(OH))(OH) blocks in the structure. The (100) reflection at 7.39° 2θ (11.95 Å) for L3 enriched by Li and 7.40° 2θ (11.94 Å) for the L3-Na form.

### 3.5. Single-Crystal X-ray Diffraction

The kukisvumite crystal structure [59] is based upon a three-dimensional framework consisting of titanosilicate layered blocks linked by ZnO_4_ tetrahedra (Figure 9a). The titanosilicate blocks are 1.2 nm thick and consist of chains of edge-sharing TiO_6_ octahedra and pyroxene-like [Si_2_O_6_] chains of corner-sharing SiO_4_ tetrahedra. The chains of the two types are linked together by sharing common O atoms. Zn^2+^ cations are located between the titanosilicate blocks and provide their linkage into a three-dimensional octahedral–tetrahedral framework that contains one-dimensional channels with a crystallographic free diameter of 4.0 × 4.8 Å^2^. The channels are parallel to [001] and are occupied by Na^+^ cations and H_2_O molecules. In addition to the intra-channel Na sites, the structure of kukisvumite contains additional Na sites in the interior of titanosilicate blocks. All Na sites are octahedrally coordinated by O atoms.

The crystal structure of lintisite [60] (Figure 10a) is closely related to that of kukisvumite and contains the same type of titanosilicate blocks. However, their linkage into a three-dimensional framework is achieved via the insertion of Li^+^ cations into the interlayer, which adopts a tetrahedral coordination. The LiO_4_ tetrahedra link titanosilicate blocks along [100] to create structural channels occupied by Na^+^ cations and H_2_O molecules.

The crystal structures of kukisvumite and lintisite could be considered as related via the Zn^2+^–2Li^+^ substitution mechanism. However, there are some principal differences between the two structures that do not allow such a straightforward comparison. As mentioned above, the titanosilicate blocks contain chains of edge-sharing TiO_6_ octahedra. These chains may have two possible orientations related by 180° rotation (Figure 11a,b), and are denoted as “+” and “−”.

The octahedral chains within one titanosilicate block are always in the same orientation, whereas orientations of the chains in adjacent blocks may be different. In the crystal structure of kukisvumite, orientations of chains in adjacent blocks are different so that blocks with positive and negative orientations alternate along the *a* axis (Figure 11c). In contrast, all octahedral chains in the crystal structure of lintisite possess the same orientation (Figure 11d). The difference between the orientations of octahedral chains (and therefore of the whole titanosilicate blocks) in kukisvumite and lintisite does not allow their consideration as chemically different but structurally identical materials; this has direct consequences for the structures of their derivative compounds described herein.

The crystal structures of K3 and L3 are shown in Figure 9b and Figure 10b, respectively. The SCSC transformation is induced by acid treatment results in leaching of all the cations, except for Ti, Si and H. In particular, leaching of the interlayer Zn^2+^ and Li^+^ cations leads to the shrinkage of the *a* unit-cell parameter and to shifting of the adjacent titanosilicate blocks relative to each other. Figure 12 shows the positions of the adjacent titanosilicate blocks in terms of the portions of the blocks consisting of a single layer of octahedral and tetrahedral chains in kukisvumite (Figure 12a,c) and K3 (Figure 12b,d). Figure 12e demonstrates that structural reconstruction involves a shift of the adjacent layers by the (*b* + *c*)/2 vector, i.e., by ~5 Å. This shift results in a much closer packing of the adjacent titanosilicate blocks due to the removal of cations and H_2_O molecules from the space between the blocks. The most remarkable feature of this transformation is that crystals do not lose their crystallinity completely, though their quality is affected considerably. Therefore, the SCSC transformation induced by the acid treatment of kukisvumite and lintisite includes the removal of low-charge cations, such as Zn^2+^, Na^+^, Li^+^ and H_2_O molecules, associated with the coordinated shifts of adjacent titanosilicate blocks by 5 Å along the (*b*
**+**
*c*) vector.

### 3.6. Rieveld Refinement

A Rietveld refinement was performed on the powder diffraction patterns, to check the consistency between the single crystal and powder sample used for the properties study. The final structure models of Ti(Si_2_O_5_(OH))(OH), obtained from SC XRD, were used in the refinement utilizing RietveldToTensor software (v1.1) [1]. Pirson VII functions were used for fitting the reflection profiles. The background was described using a Chebyshev polynomial function (21st order) and the preferred orientation (direction [100]) was modeled using the March–Dollase approach. Among structural parameters, we refined the atomic coordinates and the isotropic temperature factor, which was constrained to be the same for all atoms. The final Rietveld refinement results are given in Figure 13 and Appendix A. The final refinement resulted in values *R*_wp_ = 0.064 and *R*_Bragg_ = 0.079. The results obtained from the powder diffraction data are in good agreement with those derived from the single-crystal data.

### 3.7. Raman Spectroscopy

The Raman spectra of lintisite with L3 and kukisvumite with K3 forms are shown at Figure 14a,b, respectively. The Raman data of the AM-4 family compounds have not previously been published and the assignments of the bands were made using analogy with structurally-related titanosilicates [8,47,76,77,78]. In general, the spectra of kukisvumite and lintisite are very close, and differ only by small shifts and in the intensity of different bands.

The Si_2_O_6_ chains units produce vibrations in four frequency regions. The bands at 1092 (1081), 1057, 1027 (1030), 980 (998 w), 948 (952) and 881 (889) cm^−1^ for lintisite (in brackets kukisvumite) and 1093 (1078), (1043), 1028 (1026 w), 997 w (993 w), (981 w), 982 w, 948 (948) and 882 cm^−1^ for L3 (in brackets K3) can be attributed to symmetric and asymmetric stretching vibrations related to the non-bridging Si–O bonds in SiO_4_ tetrahedra [79]. The bands at 716 (713), 690, 670 (663) and 627 (627) cm^−1^ in the lintisite (kukisvumite) spectra and the bands at 711 (708), 671 (683), 658 and 627 (622) cm^−1^ in the L3 (K3) spectra are mainly attributed to the vibrations of the bridging Si–O–Si linkages of Si_2_O_6_ chains [80]. The bands at 567 (568), 545 s (533), 496 (494 s), 474 (475 s), (468) 437 and 421 (423) cm^−1^ in the lintisite (kukisvumite) spectra and bands at 569 (563), 532 (528), 497 (494), 475 (470 s), 437 (463) and 418 cm^−1^ in the L3 (K3) spectra are related to the bending vibrations of Si–O bonds in SiO_4_ tetrahedra and different modes of stretching vibrations of Ti–O bonds in TiO_6_ octahedra [8]. The low intensity bands at 379 (382) and 349 (366) cm^−1^ and relatively intense bands at 325 s (321) and 277 (271, 279) cm^−1^ plus in the lintisite (kukisvumite) spectra and bands at 379 (376) and 351 w (362), 325 s (319), 277 s (273 s) and 260 cm^−1^ in the L3 (K3) spectra correspond to the bending vibrations of Ti-O-Si and Ti-O-Ti bonds [81,82,83]. The bands below 210 cm^−1^ belong to translational vibrations.

The spectra of lintisite and kukisvumite generally are a close fit but are a bit different. The splitting of the two intense bands at 277 and 475 cm^−1^ in lintisite into four bands at 260, 270, 460 and 470 cm^−1^ in kukisvumite spectra is related to the different modes of vibration of Ti–O bonds. This splitting may be explained by the more complex character of kukisvumite’s structure. It (Figure 11c) contains two types of chain of TiO_6_ octahedrons; this is in contrast to lintisite, which contains only one type of chain (Figure 11d). At the same time, lintisite contains two bands at 670 and 690 cm^−1^, which are related to the vibrations of the bridging Si–O–Si linkages of Si_2_O_6_ chains, whereas kukisvumite only contains a band at 663 cm^−1^. This splitting is probably related to the more distorted SiO_4_ tetrahedra in lintisite; the polyhedral volumes of the Si1 and Si2 tetrahedra in lintisite are 2.185 and 2.207 Å^3^, which compares with kukisvumite’s corresponding volumes of 2.187 and 2.191 Å^3^.

The L3 and K3 spectra are closer than lintisite and kukisvumite. At the same time, the L3 spectra retains some of the features of lintisite spectrum, whereas K3 retains the peculiarity of kukisvumite’s spectrum. The most significant differences between L3 and K3 spectra include the splitting of bands at 475 and 277 cm^−1^ into L3 to bands at 470, 463, 273 and 260 cm^−1^ in the K3 spectrum. It seems that this spectrum features are related to its inheriting two layers of titanosilicate blocks in K3 and one layer in L3, from kukisvumite and lintisite, respectively. It is worth noting the presence of two weak peaks at 1026 and 1043 cm^−1^ in the K3 spectrum, in contrast to a band at 1028 cm^−1^ in the L3 spectrum, which is related to asymmetric stretching vibrations of Si–O bonds. The increase of number bands in K3 compares with the L3 spectra, which is related to the Si–O bonds and is probably connected with an increase in number of independent tetrahedral sites from two in L3 to four in K3.

The most dramatic changes in the Raman spectra connected with kukisvumite-K3 and lintisite-L3 transformations in the range of the H–O–H bending vibrations involve intensity increasing of the band at 3236 (3225) and 3588 cm^−1^, related to O–H vibrations in the hydroxyl group, and at 3355 (3370) cm^−1^, related to H_2_O for L3 (K3 in brackets). The increase in the hydrogen bonds’ interaction strengths leads to the increasing intensity of bands in the O–H bending vibrations region and to small changes in the Si–O vibrations connected with the protonation of non-bridging Si–O bonds in SiO_4_ tetrahedra and O atoms in TiO_6_ octahedra in the structure of K3.

### 3.8. Theoretical Calculations of the Ion-Conductive Properties for the Lintisite Group Compounds

Despite the difference in the arrangement of the chains of titanium octahedra (Figure 11a,b), the crystal structures of lintisite and kukisvumite have the same layered topology. The tiling model of the framework was constructed in order to determine the location of structural cavities with “cross-linking” (Li and Zn) atoms included into the framework. The 3D framework in lintisite has the composition Li_2_Ti_4_Si_8_O_8_. The natural tiling for this framework consists of three type of cavities (tiles) and has the formula: 2t-hes + [4^2^.5^4^.6^2^.14^2^] + [3^4^.5^4^.6^4^] (Figure 15). The smallest tile [64] is named according to the zeolite tiles classification (t-hes). The same tile has been observed previously in zeolites JBW, TON, MTT, MTW, SFE, SFN, SSO and SSY. The t-hes tile in lintisite contains an Na^+^ ion. The [3^4^.5^4^.6^4^] tile is “empty” with respect to the metal ions. The largest [4^2^.5^4^.6^2^.14^2^] tile is channel forming and has a flat «pancake» form. Its volume is not large enough to contain ions or molecules inside. The stacking of these «pancakes» corresponds to the wide channels suitable for ion-exchange properties. Such packing of the tiles allows the conclusion that the cation migration in the structures is possible along three paths: (1) the channels created by the packing of the [4^2^.5^4^.6^2^.14^2^] tiles; (2) the channels running through the t-hes tiles and (3) the channels formed by «cross-linking» cations. The first type of ion migration includes cations located inside wide channels. For the second and third types of migration, the activation energy barriers are determined using the BVSE and DFT calculations.

The calculated migration energies for the AM-4 family members, calculated using BVSE for Na^+^, Li^+^ and Zn^2+^, are given in Table 4. The lower values of the migration energy of Li ions in lintisite and Zn ions in kukisvumite indicate their possible migration. The most feasible ion-migration paths for Li^+^ in the lintisite are the 1D paths along the [001] channel system shown in Figure 16a. For lintisite and kukisvumite, the Na^+^ migration energy barrier *E_m_* is more than 1.1 eV. For the AM-4 crystal structure, the Na^+^ ions are located as in the “cross-linking” positions as well as in the intraframework sites. Thus, the migration energy barrier for 1D migration is low (0.53 eV). In total, the BVSE calculations show that the crystal structures of lintisite, kukisvumite and AM-4 are more suitable for 1D diffusion of Na^+^ and Li^+^ ions. The proton migration energies in all structures are given in Table 4. The crystal structures of kukisvumite and AM4 are suitable for 2D proton diffusion, while the crystal structure of lintisite makes 3D diffusion possible (Figure 17b). Therefore, the minerals under consideration may potentially be mixed cationic conductors.

The crystallochemical stability of the crystal structures can be estimated by determining the global instability index (*GII*) [69]. Typically, stable structures have *GII* values of less than 0.3, but this value can be slightly overestimated for disordered structures [84]. All minerals met the stability criterion *GII*.

The ionic conductivity for cations and protons was also calculated using KMC modeling. The main results of the KMC simulations of the Li^+^ and Zn^2+^ ions are presented in Appendix A and Figure 17a,b. The highest cationic conductivity in the temperature range of 300–800 K is observed for kukusvumite, exceeding other minerals by three orders of magnitude. Proton conductivity is possible in the range 300–500 K, since the conductivity ceases to change with the increasing temperature, according to the KMC simulation. The highest proton conductivity is characterized for AM-4.

We found a dependence of the distance between protons involved in charge transfer on the conductivity order in the structures (Figure 18). It can be seen that the highest value of proton conductivity is characterized for AM-4, in which the shortest distances between protons are observed. Conversely, lintisite, which has the lowest values of proton conductivity, has the greatest distance between proton carriers.

The Li sites in the Li-exchanged form of AM-4 have not been determined, and two different models were used for the DFT modelling of the cationic conductivity. The parameters of VASP optimization are given in Appendix A. The Li^+^ ions may occupy the positions of «cross-linking» cations (AM-4-I with the composition Li_2_Na_4_Ti_4_Si_8_O_27_) or take positions inside the framework (AM-4-I with the composition Li_2_Na_4_Ti_4_Si_8_O_27_).

The *E_m_* values for the diffusion of Li^+^ and Na^+^ ions were calculated for both models using the NEB method (Table 5, Figure 19). The energy barrier for Li^+^ ions along the channel formed by «cross-linking» positions is twice as high as the corresponding barrier for Na^+^. In contrast, the energy barrier for Li^+^ inside the framework is lower than the corresponding barrier for Na^+^; this can be explained by the fact that Li^+^ passes through the faces of the t-hes tiles easily, but that the ion migration of a small cation in the presence of large pores becomes more difficult, due to the interaction between Li^+^ and the framework O^2^ ions.

### 3.9. Conductivity of the AM-4:Na and AM-4:Li Modifications

In general, the temperature dependences of electrical conductivity for AM-4 (Figure 20) are characteristic for conducting materials [85]. For the AM-4 sample with graphite electrodes (Sample 1—the red curves in Figure 16 are solid at the constant frequency of 1 kHz, dashed at a short-term application of direct current (DC), *U*_meas_ = 0.12 V, the green curves correspond to the same sample in the cooling mode), the deviation from the linearity at temperatures below 150 °C is probably due to the absorption of water from air, which was confirmed by the KMC calculations. After cooling, the sample was heated to 500 °C; a decrease in conductivity was observed while the sample was in the range from 250 to 500 °C, due to the loss of the absorbed water. The AM-4 sample with silver-containing conductive paste (Sample 2—blue curves) exhibits conductivity below the sensitivity limits of the equipment used up to 250 °C. The AM-4:Li sample with silver-containing conductive paste (Sample 3—purple curves) had a conductivity of about 10^−5^ S/cm at 400 °C. It should be noted that the order of conductivity is consistent with the calculated values within the KMC approach.

Thus, at low temperature ranges, the electrical conductivity consists of two contributions by protonic and cationic ions, since a sharp increase in temperature is observed from 150 °C onwards.

The activation energy was also calculated for a number of linear curves. It has close values of 0.66–0.69 eV (fluctuations within the measurement error). The values of the experimental activation energies are in good agreement with the calculated migration energies from the BVSE and DFT data.

## 4. Discussion

Materials based upon crystal structures that can adapt in response to different external factors are attracting more and more attention from the material science community [86,87,88]. The flexibility of crystal structures is used in many applied studies that involve magnetic materials, catalysts, ferroelectric materials and sorbents [89,90,91]. In most cases, the SCSC caused by guest sorption reaction or host–guest interactions are reminiscent with those in microtubes with ferments in biological assemblies [92,93,94,95]. New information about the SCSC transformations clarify mechanisms of structural self-assembly and post-crystallization crystal chemical adaptations [8]. One of the most interesting aspects of such transformations is the nature of high-grade structural transformation, which involves shift of structural layers and significant unit-cell volume changes.

The easy and fast SCSC transformation of kukisvumite and lintisite in acid solutions results in the formation of novel acidic titanosilicate materials, K3 and L3, with the chemical formula Ti(Si_2_O_5_(OH))(OH). The same titanosilicate blocks are also present in the structure of synthetic AM-4 material [27,49] that attracted considerable interest due to its selective cation-exchange properties [49,96]. Our experiments indicate that acid treatment of AM-4 leads to the formation of the layered structure of the L3 type, which opens a possibility for its use for the creation of a new family of layered titanosilicate materials in catalysis [13,50].

The mechanism of the SCSC transformations of the kinds kukisvumite → K3, lintisite → L3 and AM-4 → L3 are accompanied by significant unit-cell contractions of ~20% and shifts of titanosilicate layers relative to each other by ~5 Å along the (*b* + *c*)/2 vector. Despite the essential structural changes, the single-crystal nature persists because of the formation of strong hydrogen bonds between the titanosilicate blocks. The new hydrogen bonds appear in agreement with the substitution schemes Li^+^ + O^2−^ → □ + OH^−^, Na^+^ + O^2−^ → □ + OH^−^ and Zn^2+^ + 2O^2−^ → □ + 2OH^−^. These chemical reactions agree with the increasing intensity of the bands corresponding to the H–O–H bending vibrations in the Raman spectrum of K3 compared to the same bands in kukisvumite (Figure 14). The general scheme of hydrogen bonding in both compounds formed by acid treatment the same and for K3 compounds is shown in Figure 21. For K3, the strongest hydrogen interactions are in between the O14–O12 and O10–O13 atom pairs with the **D**^…^**A** distances of 2.695 and 2.865 Å, respectively. The same distance for the O1–O7 pair in L3 is 2.791 Å.

It should be emphasized again that the orientations of octahedral chains in the crystal structures L3 and K3 forms (Figure 11) are different, despite having the same chemical formula, Ti(Si_2_O_5_(OH))(OH), which allows them to be considered as polymorphs.

At present, only one synthetic Na analogue, AM-4, is known for this group of minerals [26,43,44,45,46], so the obtaining of a complete Li analogue is of particular interest. Preparation of the Li-synthetic phase in sufficient quantities for practical experiments would allow us to verify our theoretical results regarding its potential electrical conductivity. However, at present, we have been unable to prepare the Li-AM-4 phase in its pure form.

The results of the modeling of the temperature dependence of the electrical conductivity of AM-4 type compounds using the KMC method agree well with the experimental data obtained from the samples with silver electrodes (after calcination at 550 °C). This indicates the applicability of this method for predicting the electrical conductivity of the materials in question. The theoretical data predict the highest level of Li^+^/Zn^2+^-ion mobility for kukisvumite. Due to the presence of several cations (Na^+^/Li^+^, Na^+^/Zn^2+^), as well as of protons, these structures are mixed cationic conductors, which can be used in various electrochemical energy sources.

## Figures and Tables

**Figure 1 materials-17-00111-f001:**
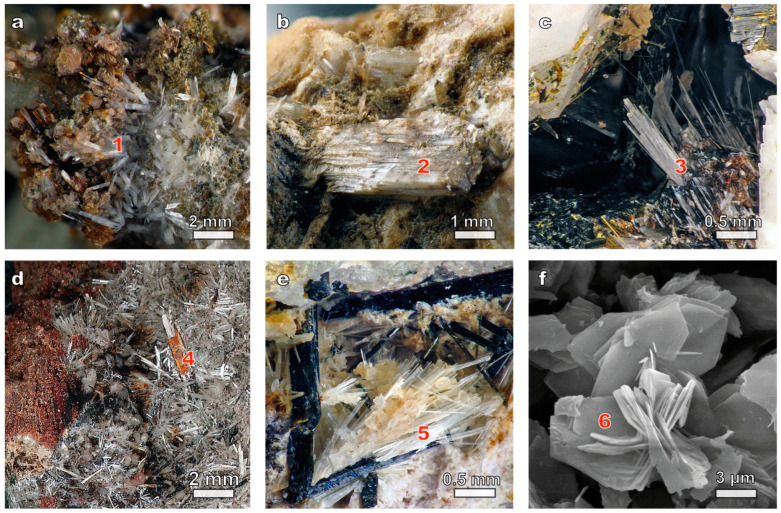
Minerals and mineral-mimetic materials belonging to the AM-4 family: (**a**) kukisvumite (1); (**b**) punkaruaivite (2); (**c**) lintisite (3); (**d**) manganokukisvumite (4); (**e**) eliseevite (5); (**f**) synthetic AM-4 (6).

**Figure 2 materials-17-00111-f002:**
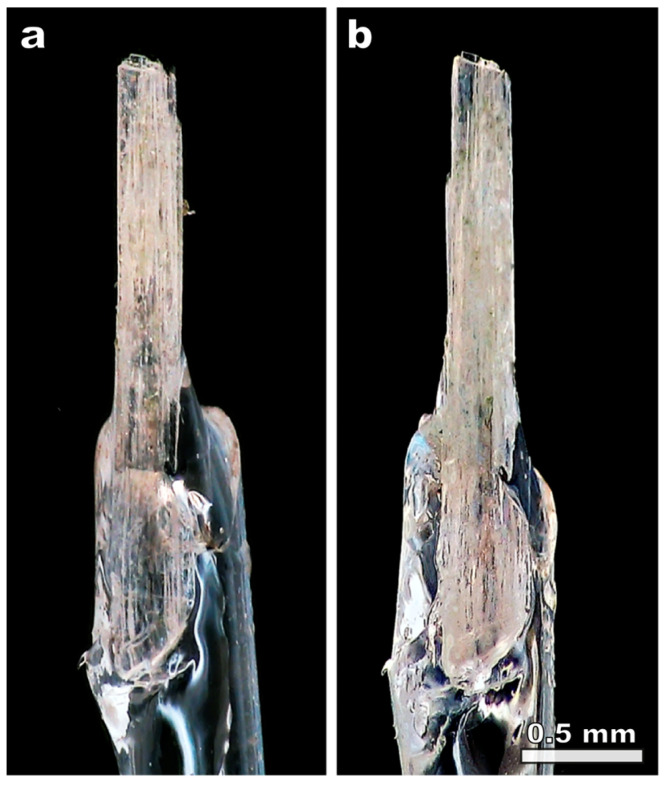
The initial crystal of kukisvumite (**a**) and the same crystal after 0.5 HCl treatment (**b**).

**Figure 3 materials-17-00111-f003:**
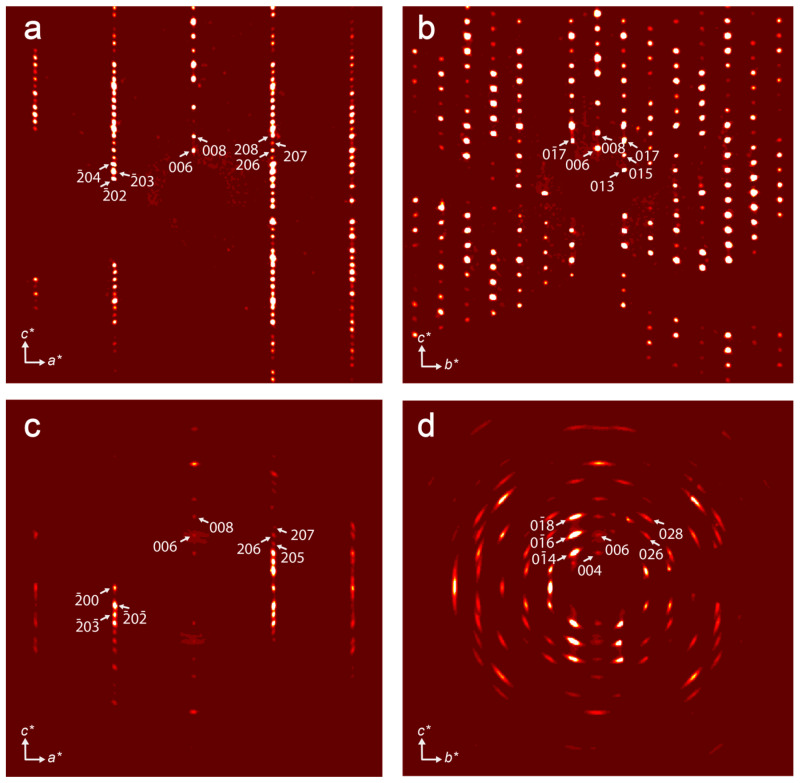
Reconstructed sections of reciprocal space obtained for (*h*0*l*) and (0*kl*) sections for kukisvumite (**a**,**b**) and K3 form (**c**,**d**).

**Figure 4 materials-17-00111-f004:**
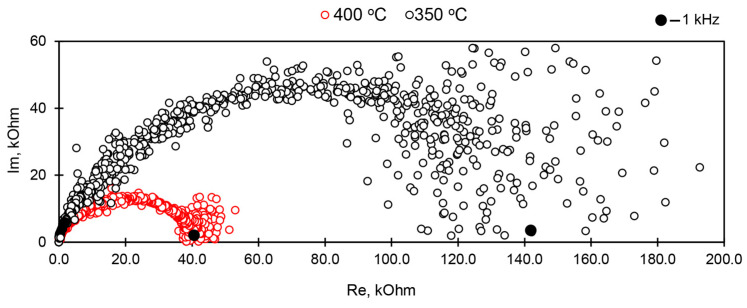
Hodographs of impedance measured at AM-4 sample with silver electrodes after calcination at 550 °C. Black markers indicate points corresponding to the 1 kHz frequency.

**Figure 5 materials-17-00111-f005:**
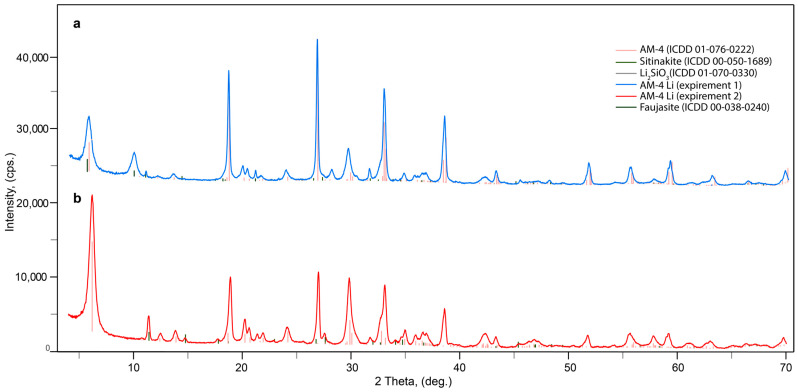
Powder X-ray diffraction patterns of Li-modification AM-4 phases with Li-metasilicate (**a**) and sitinakite (**b**) phases.

**Figure 6 materials-17-00111-f006:**
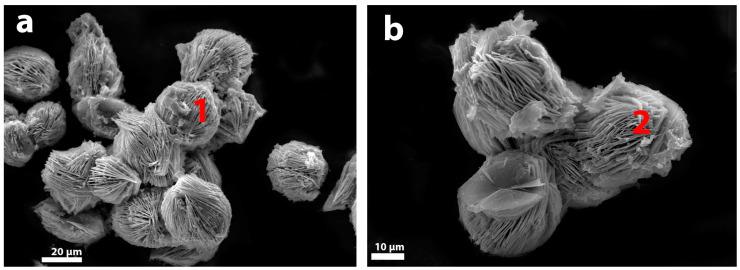
Backscattered images of (**a**) synthetic AM-4 (1) and (**b**) L3 (2) (AM-4 after HCl treatment).

**Figure 7 materials-17-00111-f007:**
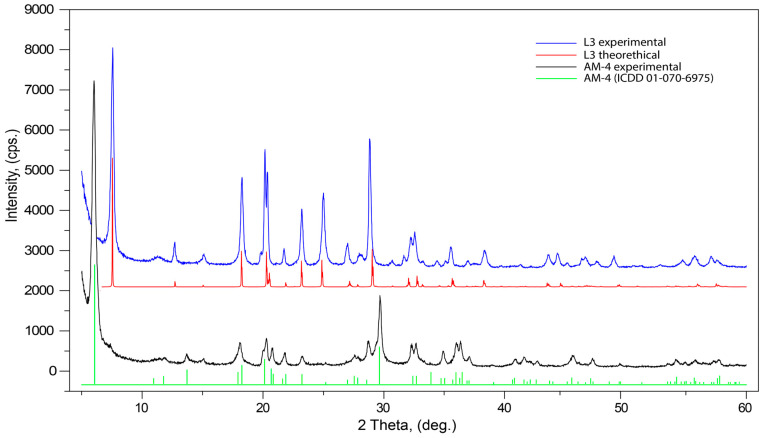
Powder X-ray diffraction patterns of initial AM-4 and L3 phases.

**Figure 8 materials-17-00111-f008:**
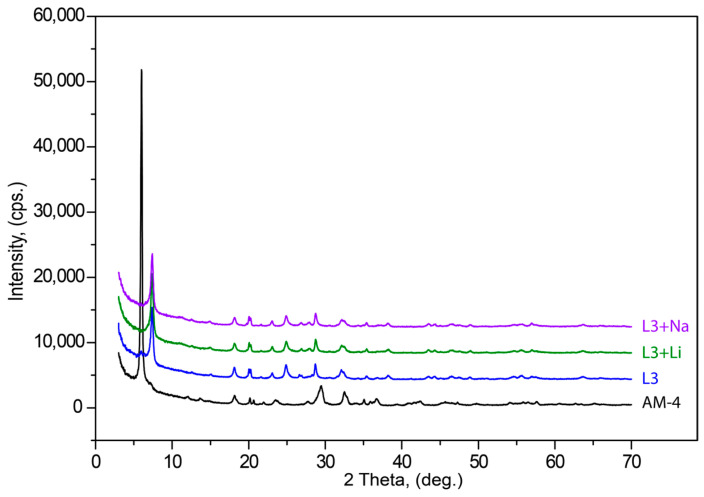
Powder X-ray diffraction patterns of initial AM-4, L3, L3 + Li and L3 + Na phases.

**Figure 9 materials-17-00111-f009:**
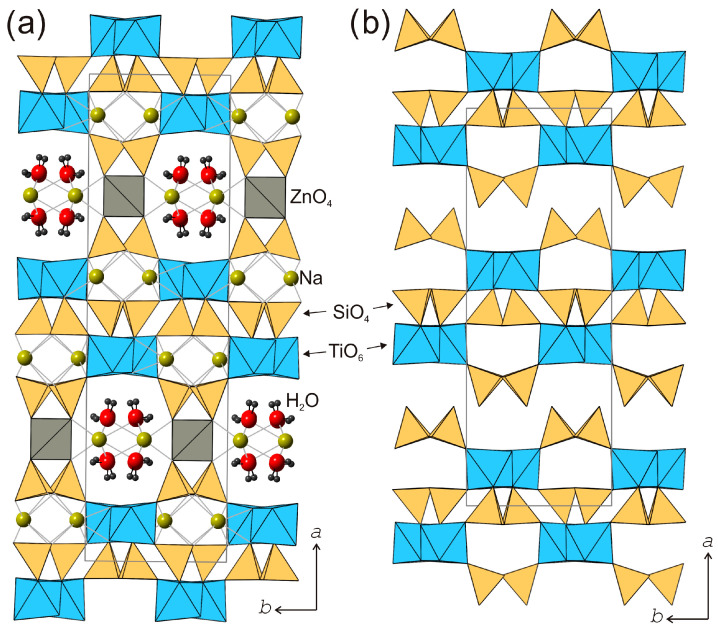
The crystal structure of kukisvumite (**a**) and its protonated modification K3 (**b**). Projection on plane (001).

**Figure 10 materials-17-00111-f010:**
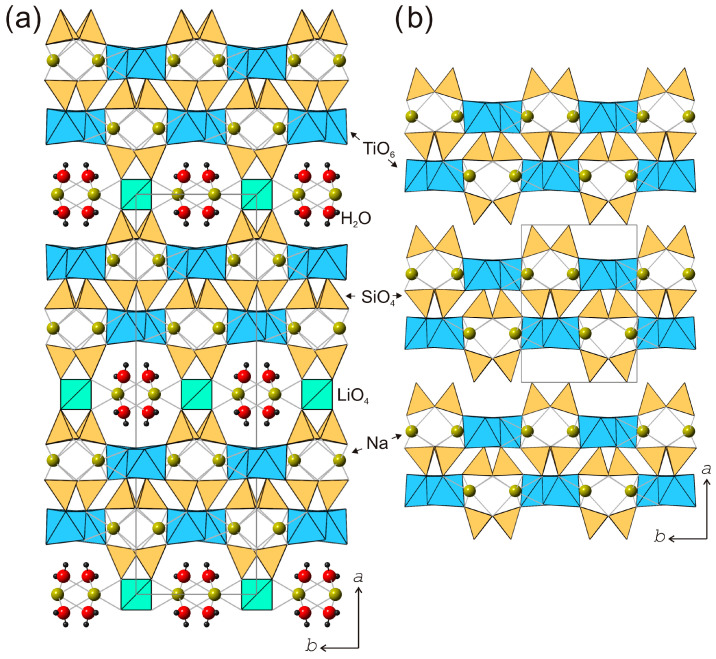
The crystal structure of lintisite (**a**) and its protonated modification L3 (**b**). Projection on plane (001).

**Figure 11 materials-17-00111-f011:**
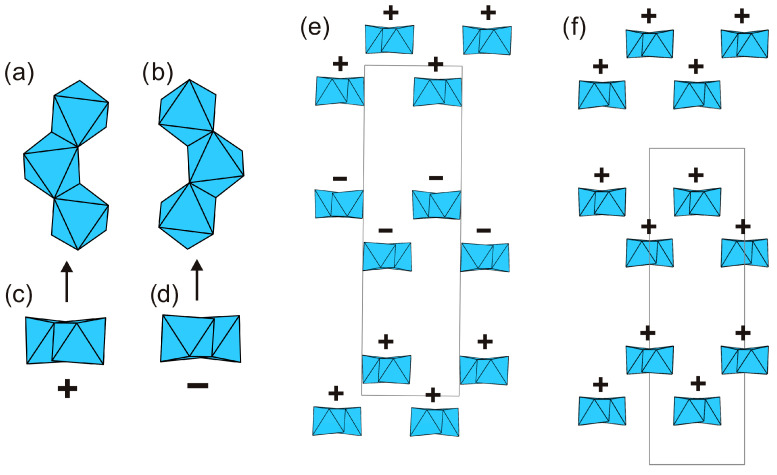
Differences between two crystal structures of kukisvumite and lintisite. (**a**,**c**), (**b**,**d**)—two type of chains TiO_6_ octahedrons. Packing of TiO_6_ chains in kukisvumite (**e**) and lintisite structure (**f**). The signs + and − means different orientation of TiO_6_ octahedron chains.

**Figure 12 materials-17-00111-f012:**
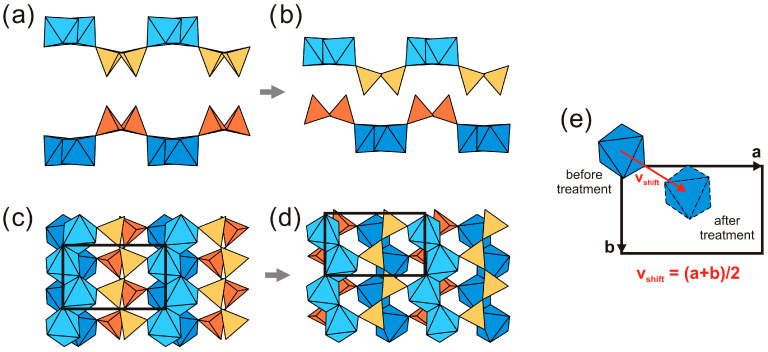
Positions of the adjacent titanosilicate in kukisvuminte (**a**) and in K3 (**b**) and structural reconstruction involving shifts of the adjacent layers for crystal structure of kukisvumite. The same fragment of unit cell in the (100) plane in kukisvuminte (**c**) and K3 (**d**) structures. Shift of TiO_6_ tetrahedra during transition (**e**).

**Figure 13 materials-17-00111-f013:**
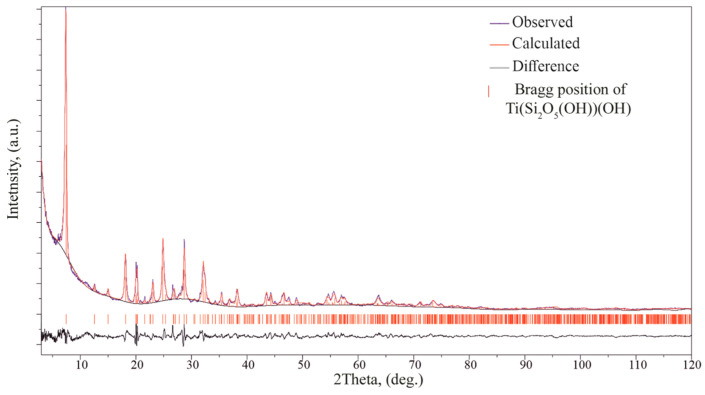
Intensity profile for the powder X-ray Rietveld refinement of Ti(Si_2_O_5_(OH))(OH). The observed and calculated profiles are represented in blue and red lines, respectively. The difference profile is plotted at the bottom. The vertical bars indicate the positions of the Bragg reflections.

**Figure 14 materials-17-00111-f014:**
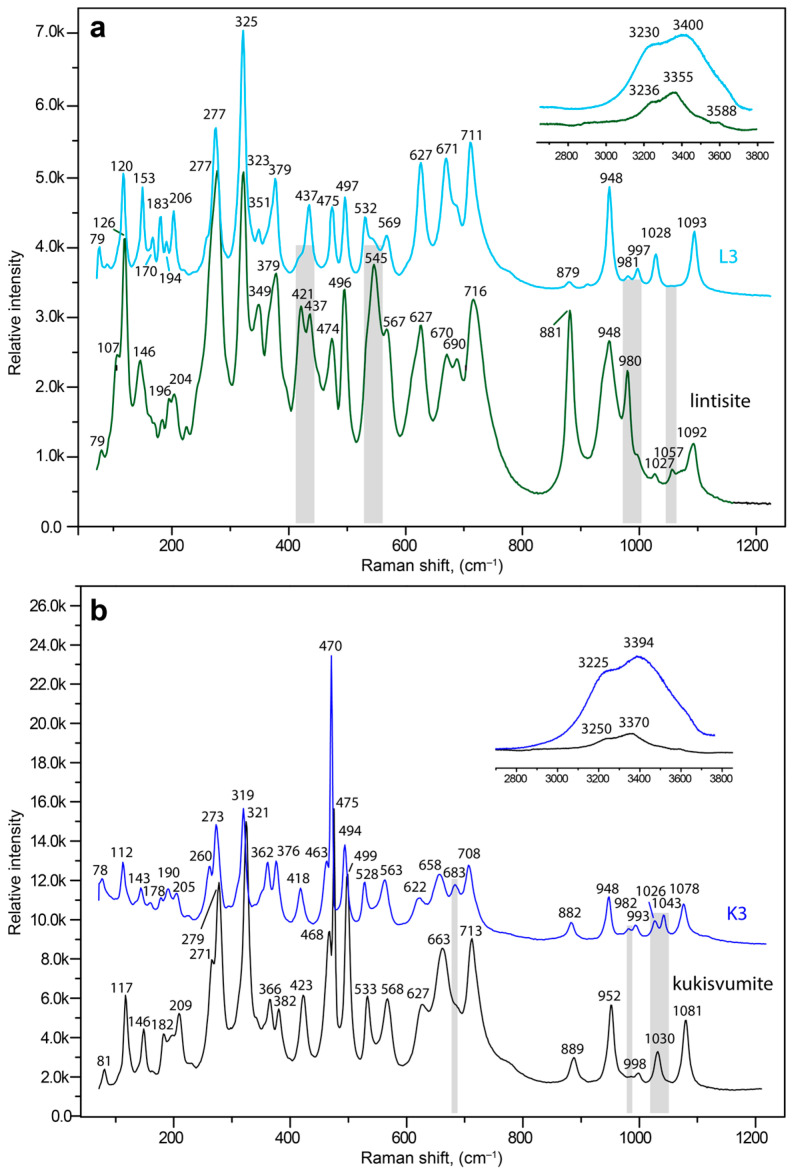
Raman spectra of pristine lintisite (green curve) and the L3 form (light blue curve) (**a**) kukisvumite (black curve) and the K3 (blue curve) form (**b**). The most significant differences in the positions or intensities of both spectra are indicated by gray lines.

**Figure 15 materials-17-00111-f015:**
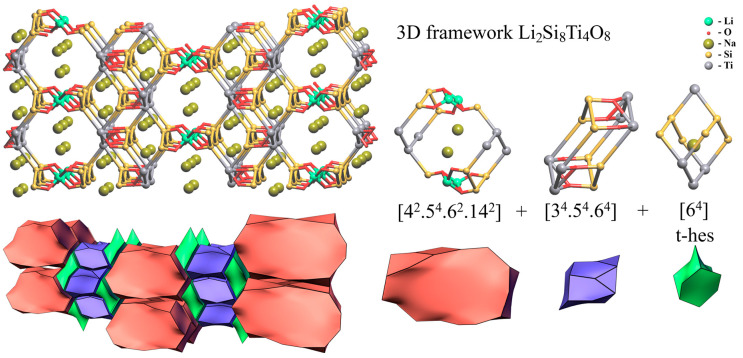
Natural tiling for lintisite structure calculated using means topological analysis.

**Figure 16 materials-17-00111-f016:**
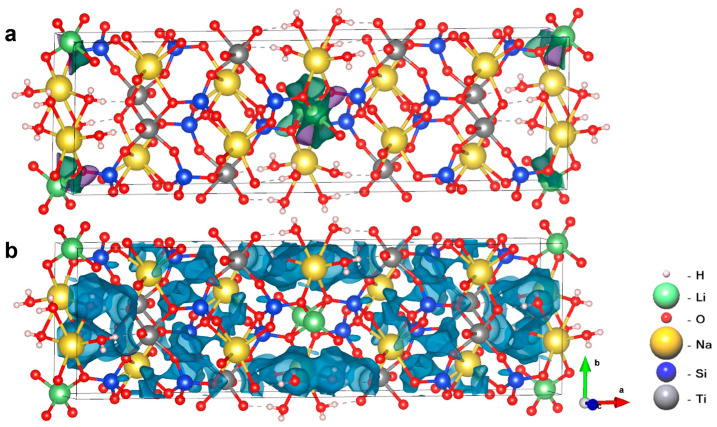
One-dimensional migration paths (green areas) of Li^+^ (**a**) and three-dimensional pathways (blue areas) of H^+^ (**b**) in the lintisite derived from the BVSE approach.

**Figure 17 materials-17-00111-f017:**
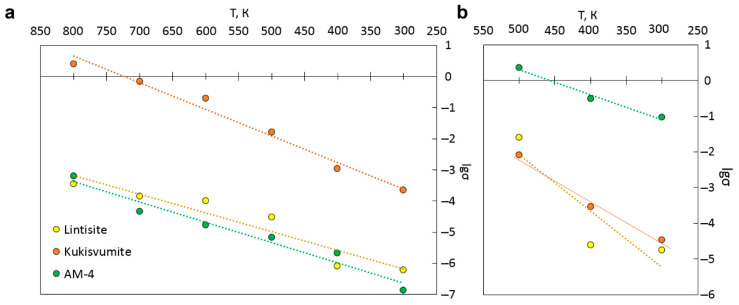
Li^+^- and Zn^2+^-ionic (**a**) and protonic (**b**) conductivities in lintisite and kukisvumite from KMC modeling.

**Figure 18 materials-17-00111-f018:**
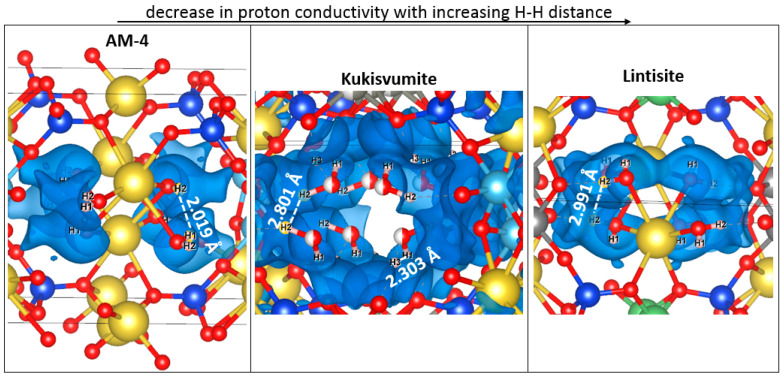
Fragments of AM-4, kukisvumite and lintisite structures, in which protons involved in charge transfer are presented in the energy isosurface, derived from the BVSE calculation with the distance between nearest ions. The atoms designation is the same with Figure 16.

**Figure 19 materials-17-00111-f019:**
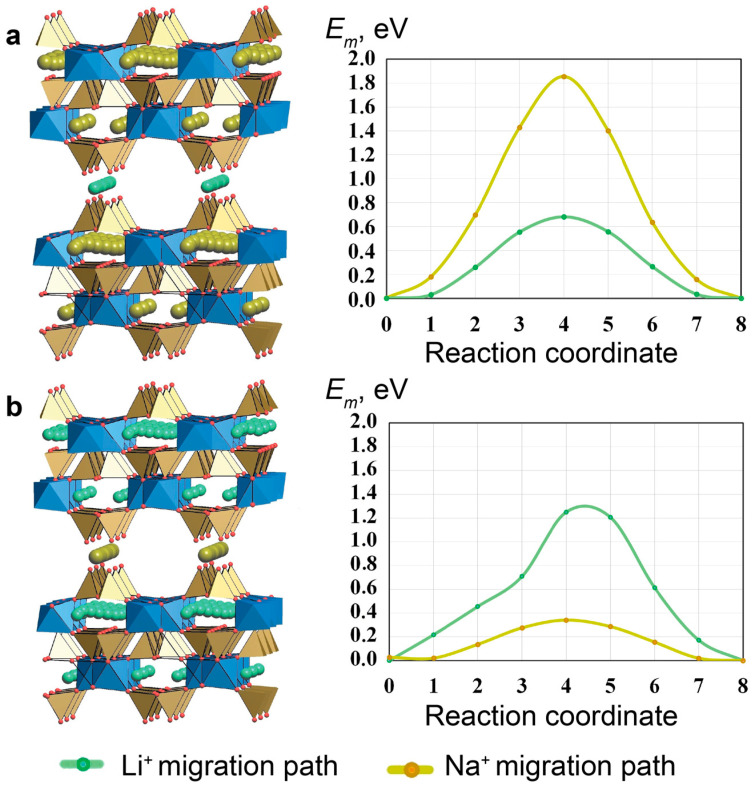
Structure and energy barriers of Li^+^- and Na^+^-ion migration for models AM-4-I (**a**) and AM-4-II (**b**). TiO_6_ octahedra filled by blue, SiO_4_ tetrahedra –brown.

**Figure 20 materials-17-00111-f020:**
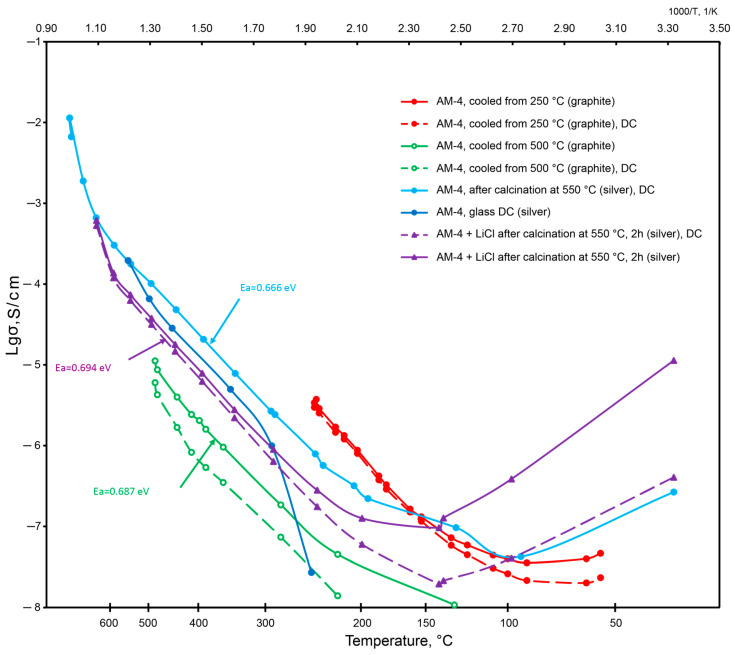
The values of the electrical conductivity of various samples depending on temperature.

**Figure 21 materials-17-00111-f021:**
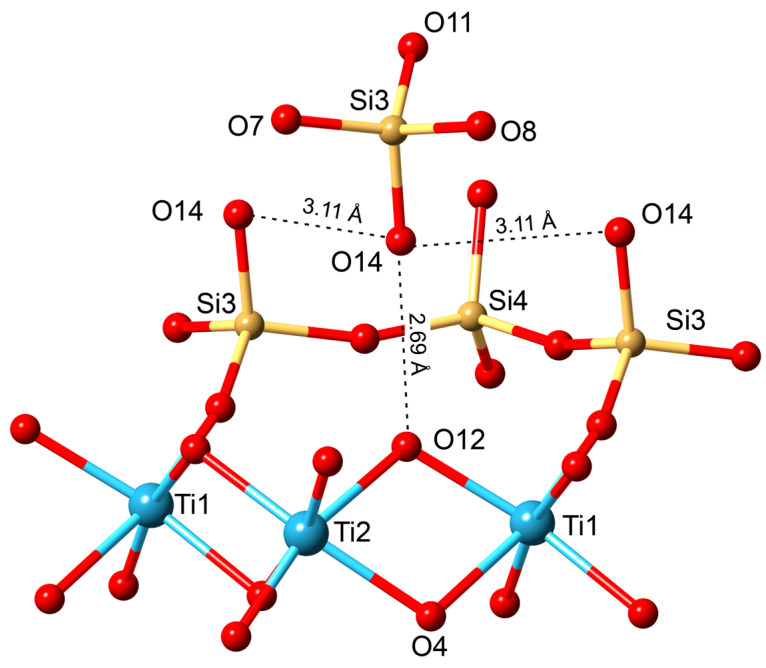
Scheme of hydrogen bond interaction in the crystal structure of K3.

**Table 1 materials-17-00111-t001:** Compounds of the AM-4 family: (mineral) name, chemical formula and citation.

Compound	Formulae	Space Group	Unit Cell Parameters	Source
Kukisvumite	Na_3_Zn_0.5_[Ti_2_Si_4_O_14_]·2H_2_O	*Pccn*	*a* = 28.889(4), *b* = 8.604(4), *c* = 5.215(3) Å	[26]
Manganokukisvumite	Na_3_Mn_0.5_[Ti_2_Si_4_O_14_]·2H_2_O	*Pccn*	*a* = 29.05(2), *b* = 8.612(6), *c* = 5.220(4) Å	[43]
Lintisite	Na_3_LiTi_2_[Si_4_O_14_]·2H_2_O	*C*2/*c*	*a* = 28.583(4), *b* = 8.600(1), *c* = 5.219(1) Å, β = 91.03(2)°	[44]
Eliseevite	Na_1.5_LiTi_2_[Si_4_O_12.5_(OH)_1.5_]·2H_2_O	*C*2/*c*	*a* = 27.48(1), *b* = 8.669(4), *c* = 5.246(2) Å, β = 90.782(8)°	[45]
Punkaruaivite	Li[Ti_2_Si_4_O_11_(OH)_3_]·H_2_O	*C*2/*c*	*a* = 26.68(1), b = 8.75(1), *c* = 5.24(1) Å, β = 91.2(2)°	[46]
AM-4	Na_3_(Na,H)[Ti_2_Si_4_O_14_]·2H_2_O	*A*2/*a*	*a* = 5.187(2), *b* = 8.582(2), *c* = 29.239(3) Å, β = 89.49(5)°	[27]

**Table 2 materials-17-00111-t002:** Crystal data, data collection and structure refinement parameters for kukisvumite, lintisite and their H-exchanged K3 and L3 forms.

Mineral/Material	Kukisvumite	K3	Lintisite	L3
Formula	Na_3_Zn_0.50_Ti_2_(Si_4_O_12_)O_2_·2H_2_O	Ti_2_(Si_4_O_5_)(OH)_2_)(OH)_2_	LiNa_3_Ti_2_(Si_4_O12)_2_O_2_·2H_2_O	Ti(Si_2_O_5_)(OH))(OH)
Temperature/K	293(2)	293(2)	293(2)	293(2)
Crystal system	orthorhombic	orthorhombic	monoclinic	monoclinic
Space group	*Pccn*	*P*2_1_2_1_2_1_	*C*2/*c*	*P*2_1_/*c*
*a*/Å	28.8905(10)	23.418(3)	28.5323(9)	11.9324(16)
*b*/Å	8.5875(3)	8.7456(8)	8.5895(3)	8.7471(11)
*c*/Å	5.2086(2)	5.2039(5)	5.2076(2)	5.2029(5)
*β*/°	90	90	90.982(3)	100.669(12)
Volume/Å^3^	1292.24(8)	1065.77(19)	1276.08(8)	533.66(11)
*Z*	4	4	4	2
*ρ*_calc_g/cm^3^	2.904	2.693	2.832	2.689
μ/mm^−1^	2.710	2.032	1.828	2.029
F(000)	1102.0	848.0	1072.0	424.0
Crystal size/mm^3^	0.22 × 0.11 × 0.08	0.1 × 0.04 × 0.03	0.17 × 0.12 × 0.1	0.2 × 0.1 × 0.09
Radiation	Mo *K*α (λ = 0.71073)			
2Θ range for data collection/°	8.464 to 66.742	6.96 to 51.988	6.392 to 66.802	6.95 to 51.988
Index ranges	−31 ≤ h ≤ 43, −12 ≤ k ≤ 12, −5 ≤ l ≤ 7	−28 ≤ h ≤ 25, −10 ≤ k ≤ 9, −5 ≤ l ≤ 6	−42 ≤ h ≤ 42, −12 ≤ k ≤ 12, −7 ≤ l ≤ 7	−14 ≤ h ≤ 14, −10 ≤ k ≤ 10, −6 ≤ l ≤ 6
Reflections collected	10,481	5222	15,644	3725
Independent reflections	2180 [*R*_int_ = 0.0303, *R*_sigma_ = 0.0251]	1982 [*R*_int_ = 0.0878, *R*_sigma_ = 0.0613]	2248 [*R*_int_ = 0.0434, *R*_sigma_ = 0.0296]	1048 [*R*_int_ = 0.0845, *R*_sigma_ = 0.0532]
Data/restraints/parameters	2180/0/133	1982/120/177	2248/0/127	1048/12/92
Goodness of fit on F^2^	1.180	2.853	1.065	1.948
Final *R* indexes [*I ≥* 2σ (*I*)]	*R*_1_ = 0.0384, w*R*_2_ = 0.1039	*R*_1_ = 0.2270, w*R*_2_ = 0.5765	*R*_1_ = 0.0331, w*R*_2_ = 0.0877	*R*_1_ = 0.1433, w*R*_2_ = 0.4223
Final *R* indexes [all data]	*R*_1_ = 0.0443, w*R*_2_ = 0.1062	*R*_1_ = 0.2354, w*R*_2_ = 0.5825	*R*_1_ = 0.0478, w*R*_2_ = 0.0937	*R*_1_ = 0.1502, w*R*_2_ = 0.4277
Largest diff. peak/hole/e Å^−3^	0.83/−0.88	6.66/−3.53	0.96/−0.61	3.43/−1.68
Flack parameter		0.5		

**Table 3 materials-17-00111-t003:** Chemical composition of kukisvumite and lintisite and their K3 and L3 counterparts obtained by acid treatment.

Constituent	Kukisvumite	K3	Lintisite	L3
in wt. %				
Li_2_O *	–	–	2.70	–
Na_2_O	16.10	0.09	14.71	0.30
SiO_2_	42.75	54.47	45.58	56.09
K_2_O	–	0.09	–	–
TiO_2_	26.86	35.08	28.83	33.99
MnO	0.74	–	–	–
FeO	0.25	0.26	0.30	0.49
ZnO	5.88	–	–	–
Nb_2_O_5_	0.86	0.37	0.52	1.49
H_2_O **	6.50	8.40	7.36	9.10 *
Sum	99.94	98.76	100.00	101.46
in formula coefficients (on the basis of Si = 4)				
Na	2.92	0.01	2.50	0.04
K	–	0.01	–	–
Li	–	–	0.95	–
Zn	0.41	–	–	–
Mn^2+^	0.06	–	–	–
Ti	1.89	1.94	1.90	1.82
Nb	0.04	0.01	0.02	0.05
Fe^3+^	0.02	0.02	0.02	0.03
Si	4.00	4.00	4.00	4.00
H	4.07	4.12	4.32	4.34
O	15.86	14.00	15.78	14.00

* calculated in accordance with SC XRD data, ** calculated in accordance with charge balance requirements.

**Table 4 materials-17-00111-t004:** Migration energies of Na^+^, Li^+^/Zn^2+^ and H^+^ ions in lintisite, kukisvumite and AM-4 derived from BVSE modeling.

Compound	Formulae	*E_m_* (Na^+^), eV			*E_m_* (Li^+^/Zn^2+^), eV			*E_m_* (H^+^), eV			*GII*	ICSD Code
		1D	2D	3D	1D	2D	3D	1D	2D	3D		
Kukisvumite	H_8_Na_6_ZnTi_4_Si_8_O_32_	1.11	1.54	>5	**0.50**	1.91	3.67	**0.39**	**0.59**	1.08	0.25	92530
AM-4	Na_8_Ti_4_Si_8_O_28_	**0.53 ***	1.15	1.15	-	-	-	**0.35**	0.85	1.20	0.29	84261
Lintisite	H_4_LiNa_3_Si_4_Ti_2_O_16_	1.49	>5	>5	**0.76**	2.43	4.12	**0.41**	**0.60**	**0.93**	0.32	-

* The most probable map dimensions and migration energies are highlighted in bold for each working ion.

**Table 5 materials-17-00111-t005:** Migration barriers for Li^+^- and Na^+^-ion diffusion in AM-4 models obtained using a DFT-NEB simulation.

Type of Atom Position	AM-4-I		AM-4-II	
	Cation	*E_m_*, eV	Cation	*E_m_*, eV
«cross-linking» position	Li^+^	0.68	Na^+^	0.34
framework position	Na^+^	1.85	Li^+^	1.25

## Data Availability

Data are contained within the article.

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
