# Peer review of "The AM-4 Family of Layered Titanosilicates: Single-Crystal-to-Single-Crystal Transformation, Synthesis and Ionic Conductivity"

_materials, 2023, doi:10.3390/ma17010111_

Round 1
Reviewer 1 Report
Comments and Suggestions for Authors
This paper describes a study of the structure and ionic conductivity of titanosilicates. It is particularly interesting in that it structurally investigates the phenomenon of the change from single crystal to single crystal upon acid treatment. The identification of the compounds and the discussion of their molecular structures are considered adequate. Therefore, it is considered to be publishable in the materials if the following points are discussed.
(1) The title should be changed.
single to single crystal transformation -> single-crystal to single-crystal transformation
(2) Additional experiments and methods should be described to determine if the compound can be reconverted to its original structure after acid treatment.
(3) Electrical conductivity should be measured in an AC electric field at various frequencies. The contribution of proton conduction should be discussed more.
Comments on the Quality of English Language-
Author Response
We are very grateful to you for the constructive comments. We tried to revise our manuscript as best as we can.
Comment:
The title should be changed: single to single crystal transformation -> single-crystal to single-crystal transformation.
Response:
Yes, we agree, the title was changed as suggested reviewer.
Comment:
Additional experiments and methods should be described to determine if the compound can be reconverted to its original structure after acid treatment.
Response:
We add additional topic 3.4 Recycling experiments with the PXRD data with AM-4, SL3, SL3 enriched by Li and Na.
Comment:
Electrical conductivity should be measured in an AC electric field at various frequencies. The contribution of proton conduction should be discussed more.
Response:
It was misunderstanding and we expanded description of measurement conditions and add new information:
The hodographs of impedance (Fig. 4) was prepared in the frequency range from 1 Hz to 1 MHz using impedance meter Z2000 in order to determine sample conductivity at alternating current in each measurement point. The frequency of 1 kHz at which measurements corresponded to the total resistance of the sample, without contribution of electrode processes was determined based on the graph. Additionally, the resistance of the samples was determined with a short-term (up to 5 s.) application of direct current, which allows evaluate the possible contribution of electrode processes by the difference of the two measurements.
Figure 4. Hodographs of impedance measured at AM-4 sample with silver electrodes after calcination at 550°C. Black markers indicate points corresponding to the 1 kHz frequency.
As for the discussion:
In low temperature intervals, the electrical conductivity consists of two contributions - proton (due to adsorbed and crystallization water) and cationic ions, since a significant increase in temperature is observed from 150 ° C. and below. The possible contribution of proton conductivity in this case can be isolated only when conducting additional studies of the samples under consideration, which were not included in the objectives of the work. This point is important for the possible practical use of the material under consideration and such measurements will be performed in the future.
However, we theoretically found a correlation between H-H distances and proton conductivity by arranging hydrogen atoms based on geometry analysis using the ToposPro software package. We found a direct relationship, which showed that as the proton distance increases, the conductivity decreases. The corresponding text and figure 18 are inserted into the manuscript.

Reviewer 2 Report
Comments and Suggestions for Authors The manuscript entitled ’The AM-4 family of layered titanosilicates: single to single crystal transformation, synthesis, ionic conductivity’ contains a lot of information, but is rather difficult to follow in the sense that it's hard to distinguish between what is actually new and what has already been done. I would recommend it for publication in Materials after revision of the text, addressing the points below:p. 6 It is stated that the (h0l) section (Fig. 3a, c) has reasonably good quality of diffraction spots. Please recheck the figures. It looks like 3 of them are showing the same section.
p. 7. The value of Rint for K3 would indicate a better final R1 value. Are there some other domains present?
p.7 Why does the preparation of K3 for Raman Spectroscopy (40min in 0.1M solution of HNO3 acid) differ from the preparation of the single-crystals for SCXRD studies (0.5M HCl for 3h)? Please provide the PXRD patterns for the K3 sample obtained both ways and compare them with the pattern generated from the crystal structure. Please add the results of Raman Spectroscopy for lintisite and L3. p. 9 please clarify: ‘Na+ + O2− → â–¡ + OH−’
p. 11 (as above) Provide the results of the PXRD studies for K3.
p. 20 Please clarify: The new hydrogen bonds appear in agreement with the substitution schemes Li+ + O2− → â–¡ + OH−, Na+ + O2− → â–¡ + OH− and Zn2+ + 2O2− → â–¡ + 2OH−.
p. 9 ‘admixture’ editorial mistake
Fig. 9 The representation is not easy to follow - to be adjusted Comments on the Quality of English Language
Moderate editing of English language required
Author Response
We are very grateful to you for the constructive comments. We tried to revise our manuscript as best as we can.
Comment:
- 6 It is stated that the (h0l) section (Fig. 3a, c) has reasonably good quality of diffraction spots. Please recheck the figures. It looks like 3 of them are showing the same section.
Response:
We agree, there is mistake in the picture, we change figure caption:
at the 3c figure axis the b axis replaced by a axis.
Comment:
- 7. The value of Rint for K3 would indicate a better final R1 value. Are there some other domains present?
Response:
Yes, the K3 really consist from domains from Pbca and P212121 symmetry. Both models are slightly different and we provide the model, that prevails. In general, the topology of both models the same.
Comment:
Why does the preparation of K3 for Raman Spectroscopy (40min in 0.1M solution of HNO3 acid) differ from the preparation of the single-crystals for SCXRD studies (0.5M HCl for 3h)? Please provide the PXRD patterns for the K3 sample obtained both ways and compare them with the pattern generated from the crystal structure. Please add the results of Raman Spectroscopy for lintisite and L3.
Response:
For the Raman spectroscopy studies, initially, we expect to demonstrate kinetics of single-crystal to single-crystal transition. And for the experiment use crystals of initial lintisite and kukisvumite. Kukisvumite were stored in in 0.1M solution of HNO3 during 1 hour. The same related to the lintisite. During this time, we reordered Raman spectra in situ. Unfortunately, the transition is permanent and we just choose the best-feet spectra. As for the lintisite- the spectra are very close to kukisvuminte one, but we add their into manuscript.
The Raman spectroscopy topic were expanded, accordingly:
The Raman spectra of lintisite with L3 and kukisvumite with K3 forms are shown at Fig. 14 a,b,respectively. The Raman data of the AM-4 family compounds have not previously been published and the assignments of the absorption bands were made by analogy with structurally related titanosilicates [8,47,76–78]. In general, the spectra of kukisvumite and lintisite are very close and differs by small shift and intensity of different bands.
The Si2O6 chains units produce vibrations in four frequency regions. The bands at 1092 (1081), 1057, 1027 (1030), 980 (998w), 948 (952) and 881 (889) cm−1 for lintisite (in brackets kukisvumite) and 1093 (1078), (1043), 1028(1026w), 997w (993w), (981w) 982w, 948 (948) and 882 cm−1 for L3 (in brackets K3) can be attributed to symmetric and asymmetric stretching vibrations related to the non-bridging Si‒O bonds in SiO4 tetrahedra [79]. The bands at 716 (713), 690, 670 (663), 627(627) cm−1 in the lintisite (kukisvumite) spectra and the bands at 711 (708), 671 (683), 658 and 627 (622) cm−1 in the L3 (K3) spectra are mainly attributed to the vibrations of the bridging Si‒O‒Si linkages of Si2O6 chains [80]. The bands at 567 (568), 545s (533), 496 (494s), 474 (475s), (468) and 437, 421 (423) cm−1 in the lintisite (kukisvumite) spectra and bands at 569 (563), 532 (528), 497 (494), 475 (470s), 437 (463) and 418 cm−1 in the L3 (K3) spectra are related to the bending vibrations of Si‒O bonds in SiO4 tetrahedra and different modes of stretching vibrations of Ti‒O bonds in TiO6 octahedra [8]. The low intensive bands at 379 (382) and 349 (366) cm−1 and relatively intense bands at 325s (321) and 277 (271, 279) cm−1 plus in the lintisite (kukisvumite) spectra and bands at 379 (376) and 351w (362), 325s (319), 277s (273s) and 260 cm−1 in the L3 (K3) spectra correspond to the bending vibrations of Ti-O-Si and Ti-O-Ti bonds [81–83]. The bands below 210 cm−1 belong to translational vibrations.
The spectra of lintisite and kukisvumite generally have close fit, but a bit different. The splitting of the intensive two bands at 277 and 475 cm−1 in lintisite into four bands at 260, 270 and 460, 470 cm−1 in kukisvumite spectra related to different modes vibration of Ti‒O bonds. Such splitting may explain by more complex character of kukisvumite structure, because it (Fig. 14 c) contains two types chains of TiO6 octahedrons, in contrast to lintisite, which contain only one type of chains (Fig. 14 d). At the same time lintisite contain two bands at 670 and 690 cm−1 related to the vibrations of the bridging Si‒O‒Si linkages of Si2O6 chains whereas kukisvumite contain only 663 cm−1 band. Probably this splitting related to the more distorted SiO4 tetrahedra in lintisite, which polyhedral volume of Si1 and Si2 tetrahedra are 2.185 and 2.207 Å3 compare with kukisvumite with corresponding volumes of 2.187 and 2.191 Å3.
Figure 14. Raman spectra of pristine kukisvumite and the K3 form. The most significant differences in the positions or intensity in both spectra indicated by gray lines.
The L3 and K3 spectra is closer than lintisite and kukisvumite. At the same time the L3 spectra retains some of the features of lintisite spectrum, whereas K3 save peculiarity of kukisvumite spectrum. The most significant differences between L3 and K3 spectra include splitting of bands at 475 and 277 cm−1 into L3 to bands at 470, 463 and 273 and 260 cm−1 in K3 spectrum. It seems that this spectrum features related to inheriting of two-layers titanosilicate blocks in K3 and one-layer in L3 from kukisvumite and lintisite, respectively. It should note presence of two weak peaks at 1026 and 1043 cm−1 in K3 spectrum in contrast to band at 1028 cm−1 in L3 spectrum related to asymmetric stretching vibrations of Si‒O bonds. Increasing of number bands in K3 compare with L3 spectra, related to the Si-O bonds probably connected with increasing number of independent tetrahedral sites from 2 in L3 to 4 in K3.
The most dramatic changes in Raman spectra connected with kukisvumite-K3 and lintisite -L3 transformations in the range of the H−O−H bending vibrations involve intensity increasing of the band at 3236 (3225) and 3588 cm−1 related to the O−H vibrations in the hydroxyl group and at 3355 (3370) cm−1 related to H2O for L3 (K3 in brackets). The increase of the hydrogen bonds interaction strengths leads to the increasing intensity bands in the O−H bending vibrations region and the small changes of the Si−O vibrations connected with the protonation of non-bridging Si‒O bonds in SiO4 tetrahedra and O atoms in TiO6 octahedra in the structure of K3.
Comment:
- 9 please clarify: ‘Na+ + O2− → â–¡ + OH−’
Response:
This type substitution includes exchange one Na site by vacancy, whereas one O atom become OH group.
Comment:
- 11 (as above) Provide the results of the PXRD studies for K3.
Response:
Due to the paucity of natural material we used only local methods for kukisvumite-K3, whereas AM-4 (analog of lintisite) is available in any quantities.
Comment:
- 20 Please clarify: The new hydrogen bonds appear in agreement with the substitution schemes Li+ + O2− → â–¡ + OH−, Na+ + O2− → â–¡ + OH− and Zn2+ + 2O2− → â–¡ + 2OH−.
Response:
When Li, Na and Zn loss the crystal structure, the part of O atoms should be protonated in order to the charge-balance requirements.
Comment:
- 9 ‘admixture’ editorial mistake
Response:
The term admixture changed by impurity
Comment:
Fig. 9 The representation is not easy to follow - to be adjusted
Response:
We made all the best to make clear character of the figures. We are glad to improve the figure quality if reviewer make specific comments for this.

Reviewer 3 Report
Comments and Suggestions for Authors
The authors presented the crystal structures of two materials (K3 and L3) obtained after acidic treatment of natural kukisvumite and lintisite and subsequent removal of the Zn2+, Na+ and Li+ cations. This exchange reaction was already reported, but the authors could now confirm the structure of the K3 and L3 phases by single-crystal X-ray diffraction thanks to a single-crystal-to-single-crystal transformation process. XRPD, elemental analysis, Raman spectroscopy further demonstrates the transformation. The acidic treatment seems to strongly damage the crystallinity of the natural compounds, since the crystal structure analysis resulted in very high R1/wR2 values for K3 and L3, and to a goodness of fit parameter far from unity (but more reliable Rint values). This brings a doubt on the possible applications of such modified AM-4 materials (as claimed by the authors, citing the example of catalysis…) Anyway, these systems have been fully studied, with the mechanism of migration of cations explored theoretically and ion conductivity measurements. I therefore recommend the publication of the manuscript in Materials.
Author Response
We are very grateful to you for the constructive comments. We tried to revise our manuscript as best as we can.
Comment:
The authors presented the crystal structures of two materials (K3 and L3) obtained after acidic treatment of natural kukisvumite and lintisite and subsequent removal of the Zn2+, Na+ and Li+ cations. This exchange reaction was already reported, but the authors could now confirm the structure of the K3 and L3 phases by single-crystal X-ray diffraction thanks to a single-crystal-to-single-crystal transformation process. XRPD, elemental analysis, Raman spectroscopy further demonstrates the transformation. The acidic treatment seems to strongly damage the crystallinity of the natural compounds, since the crystal structure analysis resulted in very high R1/wR2 values for K3 and L3, and to a goodness of fit parameter far from unity (but more reliable Rint values). This brings a doubt on the possible applications of such modified AM-4 materials (as claimed by the authors, citing the example of catalysis…) Anyway, these systems have been fully studied, with the mechanism of migration of cations explored theoretically and ion conductivity measurements. I therefore recommend the publication of the manuscript in Materials.
Response:
Authors are grateful to reviewer for the comment. The application of synthetic AM-4 in the field of catalysis was confirmed experimentally and represented in three published works:
1)Timofeeva, M.N.; Kalashnikova, G.O.; Shefer, K.I.; Mel’gunova, E.A.; Panchenko, V.N.; Nikolaev, A.I.; Gil, A. Effect of the Acid Activation on a Layered Titanosilicate AM-4: The Fine-Tuning of Structural and Physicochemical Properties. Appl. Clay Sci. 2020, 186, 105445, doi:10.
2) Kalashnikova, G.O.; Zhitova, E.S.; Selivanova, E.A.; Pakhomovsky, Y.A.; Yakovenchuk, V.N.; Ivanyuk, G.Y.; Kasikov, A.G.; Drogobuzhskaya, S.V.; Elizarova, I.R.; Kiselev, Y.G.; et al. The New Method for Obtaining Titanosilicate AM-4 and Its Decationated Form: Crystal Chemistry, Properties and Advanced Areas of Application. Microporous Mesoporous Mater. 2021, 313, 110787, doi:10.1016/j.micromeso.2020.1107871016/j.clay.2020.105445
3)Timofeeva, M.N.; Lukoyanov, I.A.; Kalashnikova, G.O.; Panchenko, V.N.; Shefer, К.I.; Yu Gerasimov, E.; Mel’gunov, M.S. Synthesis of Glycidol via Transesterification Glycerol with Dimethylcarbonate in the Presence of Composites Based on a Layered Titanosilicate AM-4 and ZIF-8. Mol. Catal. 2023, 539, 113014, doi:10.1016/j.mcat.2023.113014

Reviewer 4 Report
Comments and Suggestions for Authors
This manuscript describes the characterization of the structural transition and physical properties that occurs when the mineralogical species kukisvumite and linitisite are subject to acidic treatments to remove the alkali metal ions. The topic of the study is very interesting, but it is difficult to recommend the work for publication since the key results (the structure, and the single crystal to single crystal transformation) are supported by the weakest data in the manuscript. Indeed, the authors themselves acknowledge that the crystals were low quality, and that the structure refinement was far from perfect so that they could decipher basic structural features only. In my view, additional experimentation is necessary to support their hypotheses, and the work should be rejected in its current form. Please see the following comments.
Major points:
1. The biggest issue is the quality of the data for the K3 and L3 structures. Unfortunately, these structures provide the key to interpreting the rest of the study. As the authors themselves note (line 186-187), the acid-treated kukisvumite crystals "lost their crystallinity very significantly." Or, at least, they lost their single crystallinity significantly. But a key point in the title and throughout the manuscript is the nature of the single crystal to single crystal phase transition. So the two are at odds.
2. As a follow up to the low quality of the structures of K3 and L3, R1 and wR2 values are EXTREMELY high - to the point where one has to ask how reliable or correct even is the model. The goodness of fit values are way out of the normal range, so there is difficulty generating a reliable weighting scheme. The P2(1)2(1)2(1) structure (K3) has a Flack parameter of 0.5, which is open to the possibility that the structure is instead centrosymmetric. And, there is significant residual electron density, (6.66 and 3.43 electrons for K3 and L3) which probably still needs to be accounted for in some way. Is this indicating incomplete alkali metal removal, disorder, or some other feature? If it is due to polycrystallinity, can this be accounted for using Cell_Now and TWINABS, or an appropriate twin law? Unfortunately, the .cifs provided as supplementary information do not include extractable .hkl and .res data, so it is difficult to check on any of the number of possibilities as to why the refinement qualities are so low. But certainly wR2 values above 40-50% are outside of the range usually considered for publication of reliable results.
3. So how does one provide sufficient support to make the key structural distinctions in the article? I suggest a series of "control" experiments where the protonated modifications are synthesized directly. This could allow the authors to grow crystals of sufficient quality for a high quality X-ray refinement and to be able to describe the structure of the protonated modifications K3-syn and L3-syn with certainty. Then, that could provide support that the models they derived by treating the mineralogical samples are correct, despite their lower data quality. In any event, it would be a way to use high quality data to make the key arguments in the paper, rather than relying on low quality data.
4. Another thing that was curious, was that the authors provide the characterization for only certain samples but not others. For example, PXRD is provided for L3, but not K3. Raman spectra are discussed for K3 but not L3.
Minor points:
5. The significant peak in the blue PXRD pattern of Figure 4 does not seem to be addressed by any of the impurity phases or unreacted starting material phases mentioned by the authors.
6. Structural characterization of the Li:AM-4 sample would be highly desirable, since the authors mention that the Li sites in the Li-exchanged form of AM-4 have not been determined. It is further unclear whether we are dealing with the kukisvumite, K3, lintisite, or L3 structure types.
7. Table 2 should include the chemical formulas of the refinements.
8. AM-4 is used in that abbreviated form several times (including in the abstract, title, and introduction) before it is defined by a chemical formula. And even then, that definition only occurs in Table 1.
9. Table 1 could include additional useful information, including the space group and lattice parameters of those compounds. This is particularly useful to the current study, since lintisite and kukisvumite apparently have different structures, so one wonders how much structural variation there may be among the other members of the mineralogical family that are listed.
10. Surely the H-atom positions in K3 and L3 could not be determined with any certainty given the data quality. Figure 17 is highly speculative in this regard. Again, significant conclusions are being drawn on the basis of low quality structure refinements.
11. Although the structures of kukisvumite and lintisite are nicely refined in this work, as the authors note, their structures are already established in the literature. Including them in the current study is fine for the purposes of comparison, but they do not add enough novelty to the study to overcome the lack of high quality data for K3 and L3.
Comments on the Quality of English LanguageEnglish language was easy to understand - only a few places that need minor grammatical corrections.
Author Response
We are very grateful to you for the constructive comments. We tried to revise our manuscript as best as we can.
Comment:
- The biggest issue is the quality of the data for the K3 and L3 structures. Unfortunately, these structures provide the key to interpreting the rest of the study. As the authors themselves note (line 186-187), the acid-treated kukisvumite crystals "lost their crystallinity very significantly." Or, at least, they lost their single crystallinity significantly. But a key point in the title and throughout the manuscript is the nature of the single crystal to single crystal phase transition. So the two are at odds.
Response:
Authors are grateful to reviewer for the comment. In general, we agree with the low-quality SC XRD data connected with the partially loss of the crystallinity due to single crystal to single crystal phase transition. Nevertheless, the crystal structures of K3 and L3 were obtained directly from single crystal X-ray diffraction data of the initial crystals kukisvumite and lintisite stored in the acid and therefore such transition has single crystal to single crystal character. In order to improve the structural data, we made refinement of L3 crystal structure via Rietveld method, which confirmed accuracy of the model L3 crystal structure obtained by SC XRD method. At the same time paucity of natural material didn’t allow us get the same data for the K3.
Comment:
As a follow up to the low quality of the structures of K3 and L3, R1 and wR2 values are EXTREMELY high - to the point where one has to ask how reliable or correct even is the model. The goodness of fit values are way out of the normal range, so there is difficulty generating a reliable weighting scheme. The P2(1)2(1)2(1) structure (K3) has a Flack parameter of 0.5, which is open to the possibility that the structure is instead centrosymmetric. And, there is significant residual electron density, (6.66 and 3.43 electrons for K3 and L3) which probably still needs to be accounted for in some way. Is this indicating incomplete alkali metal removal, disorder, or some other feature? If it is due to polycrystallinity, can this be accounted for using Cell_Now and TWINABS, or an appropriate twin law? Unfortunately, the .cifs provided as supplementary information do not include extractable .hkl and .res data, so it is difficult to check on any of the number of possibilities as to why the refinement qualities are so low. But certainly, wR2 values above 40-50% are outside of the range usually considered for publication of reliable results.
Response:
We agree with the comment related the K3 crystal structure, it’s really consisted from intergrowths of domains with Pbca and P212121 symmetry. Both models are slightly different and we provide the model, that prevails. In general, the topology of both models the same. We check presence of the twins by TWINABS and no evidence of twining were observed. The .cif files (together with .hkl and .fcf files were deposited into CCDC under the entries No. 2266154-2266157 and can be extracted). In any case, we have refined the structure from the powder and obtained correct wR2 values (Table S13) for the powder. The residual density peaks have no physical significance and do not refer to possible positions of atoms.
Table S13. X-ray Rietveld refinement of the Ti(Si2O5(OH))(OH) structure.
|
Chemical formula |
Ti(Si2O5(OH))(OH) |
|
Mr |
432.09 |
|
Temperature (°C) |
23(2) |
|
Crystal system, space group |
Monoclinic, P21/c |
|
a, b, c (Å) |
11.92376(45), 8.72691(57), 5.20727(37) |
|
β (°) |
100.8989(94) |
|
V (Å3) |
532.08(11) |
|
Z |
12 |
|
Dx (Mg m–3) |
2.70 |
|
Radiation type |
Cu Kα |
|
Data collection |
|
|
Diffractometer |
Rigaku SmartLab SE |
|
θ-range |
1.5–120, Step size (°) 0.02 |
|
(sin θ/λ)max (Å–1) |
0.562 |
|
Refinement |
|
|
|
Rwp 0.064 |
|
|
Rp 0.066 |
|
|
RBragg 0.079 |
|
S |
1.0 |
|
No. of parameters |
53 |
Additional author Sergey N. Volkov and topics “Rieveld refinement” added into main text (in experimental and results sections
For the Rietveld refinement PXRD patterns were collected on Rigaku SmartLab SE (3 kW sealed X-ray tube, D/teX Ultra 250 silicon strip detector, vertical type θ-θ geometry, HyPix-400 (2D HPAD) detector). PXRD data were collected at room temperature in the 2θ range between 3° and 120° with a step interval of 0.02°. Rietveld refinement was performed on the powder diffraction patterns. The structure model of Ti(Si2O5(OH))(OH) was used as starting models obtained from SCXRD (without H atoms) in the refinement utilizing RietveldToTensor software [1].
A Rietveld refinement was performed on the powder diffraction patterns, to check the consistency between the single crystal and powder sample used for the properties study. The final structure model of Ti(Si2O5(OH))(OH) obtained from SC XRD were used in the refinement utilizing RietveldToTensor software [1]. Pirson VII functions was used for fitting the reflection profiles. The background was described by a Chebyshev polynomial function (21st order) and the preferred orientation (direction [100]) was modeled by the March-Dollase approach. Among structural parameters, we refined the atomic coordinates and the isotropic temperature factor, which was constrained to be the same for all atoms. The final Rietveld refinement results are given in Fig. 13 and Table S13. The final refinement resulted in values Rwp = 0.064 and RBragg = 0.079. The results obtained from the powder diffraction data are in good agreement with those derived from the single-crystal data.
Figure 13. Intensity profile for the powder X-ray Rietveld refinement of Ti(Si2O5(OH))(OH). The observed and calculated profiles are represented in blue and red lines, respectively. The difference profile is plotted at the bottom. Vertical bars indicate the positions of the Bragg reflections.
Comment:
So how does one provide sufficient support to make the key structural distinctions in the article? I suggest a series of "control" experiments where the protonated modifications are synthesized directly. This could allow the authors to grow crystals of sufficient quality for a high-quality X-ray refinement and to be able to describe the structure of the protonated modifications K3-syn and L3-syn with certainty. Then, that could provide support that the models they derived by treating the mineralogical samples are correct, despite their lower data quality. In any event, it would be a way to use high quality data to make the key arguments in the paper, rather than relying on low quality data.
Response:
Our data on the structural refinement of L3 (synthetic) using the Rietveld method fully confirm the correctness of the model obtained by X-ray diffraction studies. Unfortunately, due to the lack of natural matter, it is currently impossible to obtain similar data for K3.
It is known that the growth of single crystals of titanosilicates is extremely difficult work and in many cases it is impossible to obtain crystals suitable for X-ray structural analysis (references to sitinakite). We have been conducting experiments on AM-4 synthesis since 2014. During 9 years of research, unfortunately, it was not possible to synthesize an analog of cukiswumite. For AM-4, the largest aggregates (Fig. 1) were obtained during a 5-day experiment. Increasing the holding time does not lead to an increase in crystallite size.
Figure 1. The largest AM-4 crystals obtained during the synthesis.
Comment:
Another thing that was curious, was that the authors provide the characterization for only certain samples but not others. For example, PXRD is provided for L3, but not K3. Raman spectra are discussed for K3 but not L3.
Response:
The natural material ilintisite and kukisvumite available in amounts only for a local-methods studies (Raman, SC XRD, EDS-WDS). As requested by reviewer 1 we add Raman spectra for L3.
Comment:
The significant peak in the blue PXRD pattern of Figure 4 does not seem to be addressed by any of the impurity phases or unreacted starting material phases mentioned by the authors
Response:
We checked XRD files and add information about impurity in the text:
The impurity of zeolite with faujasite structure sometimes observed
Comment:
Structural characterization of the Li:AM-4 sample would be highly desirable, since the authors mention that the Li sites in the Li-exchanged form of AM-4 have not been determined. It is further unclear whether we are dealing with the kukisvumite, K3, lintisite, or L3 structure types.
Response:
We agree and will study Li:AM-4 sample in more detail in future works related to the ion-conductivity properties of AM-4 type compounds. However, we theoretically found a correlation between H-H distances and proton conductivity by arranging hydrogen atoms based on geometry analysis using the ToposPro software package. We found a direct relationship, which showed that as the proton distance increases, the conductivity decreases. The corresponding text and figure 18 are inserted into the manuscript.
Comment:
Table 2 should include the chemical formulas of the refinements.
Response:
We add refined formulas at Table 2.
Comment:
AM-4 is used in that abbreviated form several times (including in the abstract, title, and introduction) before it is defined by a chemical formula. And even then, that definition only occurs in Table 1.
Response:
We decipher AM-4 abbreviation in introduction (See line 62):
Aveiro-Manchester-4 (AM-4) titanosilicate
Comment:
Table 1 could include additional useful information, including the space group and lattice parameters of those compounds. This is particularly useful to the current study, since lintisite and kukisvumite apparently have different structures, so one wonders how much structural variation there may be among the other members of the mineralogical family that are listed.
Response:
We add suggested information in the Table 1 as proposed reviewer.
Comment:
Surely the H-atom positions in K3 and L3 could not be determined with any certainty given the data quality. Figure 17 is highly speculative in this regard. Again, significant conclusions are being drawn on the basis of low-quality structure refinements.
Response:
Studies of the L3 structure using the Rietveld method confirmed the correctness of our data. We believe that in the case of K3 the situation is similar and we are sure that the arrangement of O atoms in the structure is correct.
Comment:
Although the structures of kukisvumite and lintisite are nicely refined in this work, as the authors note, their structures are already established in the literature. Including them in the current study is fine for the purposes of comparison, but they do not add enough novelty to the study to overcome the lack of high-quality data for K3 and L3.
Response:
We want to emphasize especially that structural data on the K3 and L3 models have not been published anywhere before. Of course, in our earlier works the L3 scheme was given, but the details of the crystal structure of both structures are published here for the first time. The Rietveld refinement approved our structure model for L3.

Round 2
Reviewer 2 Report
Comments and Suggestions for Authors
The authors responded to all queries. Regarding Figure 9, would it be possible to present the protonated form as a fragment which closely corresponds to the representation of kukisvumite?
Comments on the Quality of English LanguageModerate editing of the English language is required.